



# The Regional Aerosol Model Intercomparison Project (RAMIP)

Laura J. Wilcox[1], Robert J. Allen[2], Bjørn H. Samset[3], Massimo A. Bollasina[4], Paul T. Griffiths[5], James M. Keeble[5], Marianne T. Lund[3], Risto Makkonen[6], Joonas Merikanto[6], Declan O'Donnell[6], David J. Paynter[7], Geeta G. Persad[8], Steven T. Rumbold[1], Toshihiko Takmeura[9], Kostas Tsigaridis[10,11], Sabine Undorf[12], and Daniel M. Westervelt[13,11]

[1]National Centre for Atmospheric Science, University of Reading, Reading, UK
[2]Department of Earth and Planetary Sciences, University of California Riverside, Riverside, CA, USA
[3]CICERO Center for International Climate Research, Oslo, Norway
[4]School of GeoSciences, University of Edinburgh, UK
[5]National Centre for Atmospheric Science, University of Cambridge, Cambridge, UK
[6]Finnish Meteorological Institute, Climate Research, Helsinki, Finland
[7]NOAA/Geophysical Fluid Dynamics Laboratory, Princeton, New Jersey, USA
[8]Department of Geological Sciences, The University of Texas at Austin, Austin, TX, USA
[9]Research Institute for Applied Mechanics, Kyushu University, Fukuoka, Japan
[10]Center for Climate Systems Research, Columbia University, New York, NY, USA
[11]NASA Goddard Institute of Space Studies, New York, NY, USA
[12]Potsdam Institute for Climate Impact Research, Potsdam, Germany
[13]Lamont-Doherty Earth Observatory, Columbia Climate School, New York, USA

**Correspondence:** Laura Wilcox (l.j.wilcox@reading.ac.uk)

**Abstract.** Changes in anthropogenic aerosol emissions have strongly contributed to global and regional trends in temperature, precipitation, and other climate characteristics, and have been one of the dominant drivers of decadal trends in Asian and African precipitation. These, and other, influences on regional climate from changes in aerosol emissions are expected to continue, and potentially strengthen, in the coming decades. However, a combination of large uncertainties in emissions
pathways, radiative forcing, and the dynamical response to forcing makes anthropogenic aerosol a key factor in the spread in near-term climate projections, particularly on regional scales, and therefore an important one to constrain. For example, in terms of future emissions pathways, the uncertainty in future global aerosol and precursor gas emissions by 2050 is as large as the total increase in emissions since 1850. In terms of aerosol effective radiative forcing, which remains the largest source of uncertainty in future climate change projections, CMIP6 models span a factor of five, from -0.3 to -1.5 W m$^{-2}$. Both of these
sources of uncertainty are exacerbated on regional scales.

The Regional Aerosol Model Intercomparison Project (RAMIP) will deliver experiments designed to quantify the role of regional aerosol emissions changes in near-term projections. This is unlike any prior MIP, where the focus has been on changes in global emissions and/or very idealized aerosol experiments. Perturbing regional emissions makes RAMIP novel from a scientific standpoint, and links the intended analyses more directly to mitigation and adaptation policy issues. From a science
perspective, there is limited information on how realistic regional aerosol emissions impact local as well as remote climate conditions. Here, RAMIP will enable an evaluation of the full range of potential influences of realistic and regionally varied aerosol emission changes on near-future climate. From the policy perspective, RAMIP addresses the burning question of how



local and remote decisions affecting emissions of aerosols influence climate change in any given region. Here, RAMIP will provide the information needed to make direct links between regional climate policies and regional climate change.

RAMIP experiments are designed to explore sensitivities to aerosol type and location, and provide improved constraints on uncertainties driven by aerosol radiative forcing and the dynamical response to aerosol changes. The core experiments will assess the effects of differences in future global and regional (East Asia, South Asia, Africa and the Middle East) aerosol emission trajectories through 2051, while optional experiments will test the nonlinear effects of varying emission location and aerosol types along this future trajectory. All experiments are based on the Shared Socioeconomic Pathways, and are intended

to be performed with sixth Climate Model Intercomparison Project (CMIP6) generation models, initialised from the CMIP6 historical experiments, to facilitate comparisons with existing projections. Requested outputs will enable analysis of the role of aerosol in near-future changes in, for example, temperature and precipitation means and extremes, storms, and air quality.

## 1   Introduction

Aerosols emitted from natural and anthropogenic sources exert strong influences on the Earth's climate. At the global mean scale, anthropogenic emissions of aerosols, such as black carbon (BC) from incomplete combustion, and of aerosol precursor gases, such as $SO_2$ that leads to the formation of sulfate particles, currently induce a net, global, annual mean cooling of around 0.4°C (Masson-Delmotte et al., 2021). Aerosols cool the climate through their interaction with radiation, and through their influence on cloud properties (Forster et al., 2021). Anthropogenic aerosols (AA) also have a wide range of direct and

indirect effects on the water and energy cycles across a range of spatio-temporal scales (Richardson et al. (2018); Samset et al. (2018a); Sand et al. (2020)), on clouds (Amiri-Farahani et al. (2017); Allen et al. (2019a); Cherian and Quaas (2020)), and on extreme weather events (Samset et al. (2018c); Samset et al. (2018b); Fan et al. (2016); Sillmann et al. (2019); Wang et al. (2020); Chen et al. (2019); Luo et al. (2020)), mediated by multiple physical mechanisms. However, while aerosol emissions are second only to greenhouse gases in contributing to anthropogenic climate change over the historical era (Forster et al.,

2021), their influences are distinct, and markedly more uncertain and spatially heterogeneous. This applies to the Effective Radiative Forcing (ERF) induced by aerosol emissions, where aerosol is the largest uncertainty in the anthropogenic forcing of climate (Forster et al., 2021), to the resulting influence on global mean surface temperature and precipitation, and, in particular, to the influence on the regional and seasonal pattern of impact-relevant climate hazards. The regional response to aerosol changes has become an increasingly active topic of research in recent years (e.g. Nordling et al. (2019); Krishnan et al. (2020);

Hari et al. (2020); Westervelt et al. (2020a); Wilcox et al. (2020); Fiedler and Putrasahan (2021); Persad (2022)). This has been motivated by recognition that a lack of understanding of regionally heterogeneous aerosol-climate effects is hampering our understanding of historical climate change, during which greenhouse gas and aerosol emissions have broadly increased in





lockstep. It also limits our confidence in future climate projections and the assessment of their impacts, as aerosol emissions are expected to rapidly decline over the coming decades.

Global anthropogenic emissions of a range of aerosol species, and the resulting aerosol optical depth (AOD), increased through most the the 20th century, levelled off in the early 1980s, and have recently begun to decline (Turnock et al. (2020); Dittus et al. (2020), Quaas et al. (2022)). The geographical distribution of emissions, and AOD, has continued to evolve since 1980, with a gradual shift of the core emission region from Europe and the US to Southern and Eastern Asia (Myhre et al. (2017a); Fiedler and Putrasahan (2021)), with a possible shift to Africa in the future (Lund et al., 2019). In the last decade
Chinese $SO_2$ emissions have been markedly reduced, while emissions of both $SO_2$ and BC from India have increased, leading to a dipole change in AOD over South and East Asia (Samset et al., 2019). For the coming decades, the Shared Socioeconomic Pathways (SSPs) used e.g. by the Intergovernmental Panel on Climate Change (IPCC) in their 6th Assessment Report (AR6) project a wide range of possible trajectories of AA emissions from different regions, depending on national and international air quality policies, the pace of energy and transport technologies transitioning away from fossil fuel combustion, and other
factors (Rao et al. (2017); Hoesly et al. (2018); Lund et al. (2019)).

Recent literature has documented how both global and regional climate are highly sensitive to regional aerosol emissions and their rates of change, with aerosol influencing the atmosphere (Undorf et al. (2018); Westervelt et al. (2018); Tang et al. (2018); Wilcox et al. (2019); Luo et al. (2020); Persad (2022)), and ocean heat uptake and circulation (Ma et al. (2020); Menary et al. (2020); Hassan et al. (2021); Robson et al. (2022); Hassan et al. (2022)). Aerosol-climate interactions are also strongly
dependent on the physical and chemical properties of the aerosols themselves, notably whether or not they absorb shortwave radiation through the atmospheric column (like BC) or predominantly scatter it (like e.g. sulfate). Globally, the hydrological sensitivity to aerosol emissions (precipitation change per unit of global mean temperature change) is about twice as high for aerosols as for greenhouse gases (Kloster et al. (2010); Salzmann (2016); Samset et al. (2016); Samset et al. (2018a)). This alone signals potential strong regional changes in precipitation if strong air pollution reduction measures are implemented in current
high-emission regions. The main reason for the difference between the precipitation response to aerosols and greenhouse gases is the lack of absorption for sulfate aerosols. Atmospheric absorption induces rapid adjustments to the atmospheric energy balance that inhibits precipitation formation. For greenhouse gases, this opposes the increase in precipitation associated with surface warming.

Aerosols have been shown to influence relevant climate phenomena local to the emission sources, such as monsoons (West-
ervelt et al. (2020a); Xie et al. (2020)), as well as to generate climate anomalies far downstream of the aerosol source regions via teleconnections (Smith et al. (2016); Undorf et al. (2018); Wilcox et al. (2019); Amiri-Farahani et al. (2020); Merikanto et al. (2021)). For precipitation, the response of the Asian summer monsoon to aerosols has been found to be particularly strong (Levy et al. (2013); Westervelt et al. (2015); Acosta Navarro et al. (2017); Bartlett et al. (2018); Samset et al. (2018b)). Aerosols also affect global circulation patterns and the interhemispheric temperature contrast, by affecting the albedo of the Northern
Hemisphere (NH) more strongly (primarily via aerosol-cloud interactions, e.g. Wilcox et al. (2013)). This has been linked to changes in the tropical rain belt (Allen et al. (2014); Allen et al. (2015); Allen and Ajoku (2016); Westervelt et al. (2017)), and the global (Polson et al. (2014); Shonk et al. (2020)) and regional monsoons (Hari et al. (2020); Westervelt et al. (2020b); Xie





et al. (2020)). Aerosols have also been found to generate teleconnections from the tropics to the NH mid-latitudes, affecting extratropical temperature and precipitation patterns and variability, and storm tracks (Ming et al. (2011); Wilcox et al. (2019);

Allen and Zhao (2022)). The frequency and intensity of extreme events have also been shown to have different sensitivites to aerosol emissions than to greenhouse-gas induced global warming (e.g. Sillmann et al. (2019); Luo et al. (2020); Chen et al. (2019)).

This all points to a strong need for improved understanding of the role of regional aerosol-climate interactions in historical and future changes in climate hazards and risk. There are however major known limitations, uncertainties, and gaps in current

scientific knowledge related to both the interaction between aerosols and climate, and aerosol emissions and formation.

Firstly, representations of aerosol-climate interactions vary markedly between current global models (Turnock et al. (2020); Wilcox et al. (2015)). Aerosols have been rudimentarily included since the 1990s, but the detail level of this implementation has greatly increased in recent years (Wilcox et al. (2013); Ekman (2014); Chen et al. (2021)). For the 6th phase of the Coupled Model Intercomparison Project (CMIP6; Eyring et al. (2016)), most participating Earth System Models included treatment of

both anthropogenic and natural emissions; notably sulfate and black carbon aerosols, and biomass burning, dust and sea spray, respectively, with many models also treating secondary organic aerosols. Emissions of natural aerosols are climate dependent. Aerosol transport, removal and deposition is treated, as is chemical processing and ageing, direct and indirect interactions with clouds, and, in many cases, internal mixing, the mixing state describing how chemical species are mixed inside the particles, which impacts aerosol optical properties and their effectiveness as cloud condensation nuclei. Crucially, the models

also include treatment of the weather and climate effects of the radiative and microphysical aerosol interactions with clouds and precipitation. Over the historical era, CMIP6 models estimate a total aerosol Effective Radiative Forcing ranging from -0.3 to -1.5 W m$^{-2}$ (Forster et al., 2021), which is equivalent to an uncertainty in historical surface temperature change of over 1 K (Dittus et al., 2020). The majority of this uncertainty arises from the aerosol-cloud interactions (Forster et al., 2021). Consequently, the complexity of their climate interactions, and their implementation in models, introduce a major component

of the remaining scientific uncertainty in both simulations of historical climate (Wilcox et al. (2013); Ekman (2014); Zhang et al. (2021a)), and future projections (Allen, 2015).

Secondly, some aerosol emission inventories, notably for carbonaceous aerosols, are more uncertain than for greenhouse gases, both in abundance and in geographical distribution (Hoesly et al., 2018). Formation of secondary aerosols, condensational growth and coagulation, transport, removal, and ageing of aerosols are all complex processes that are difficult to model,

as are all aspects of aerosol-cloud interactions (Stevens and Feingold (2009); Boucher et al. (2013); Fan et al. (2016); Szopa et al. (2021)). The optical properties of aerosols are also not fully constrained, meaning that their radiative interactions and resulting radiative forcing also have marked uncertainties. For these reasons, aerosols were highlighted in the IPCC AR6 as a major source of uncertainty in future climate projections, as they have also been in previous IPCC reports (Myhre et al. (2013); Szopa et al. (2021)). Further, simulations of aerosol-climate interactions are also dependent on the model representa-

tion of the underlying climate, such as the geographical and temporal distribution of precipitation, monsoon dynamics, modes of variability, cloud distributions and processes (Mülmenstädt and Wilcox, 2021). Most existing literature regarding regionally heterogeneous climate responses to changing aerosol emissions has been based around idealised regional perturbations (e.g.



Dong et al. (2014); Dong et al. (2016); Westervelt et al. (2017); Liu et al. (2018); Persad and Caldeira (2018)), or drawn from simulations where global emission changes are imposed (e.g. Song et al. (2014); Guo et al. (2021); Zhang et al. (2021b)).

While this has yielded a wide range of strong, fundamental insights, these approaches also have challenges. Idealised perturbations are usually artificially very large, or applied in equilibrium simulations (e.g. Myhre et al. (2017b); Westervelt et al. (2020a)), which makes the signal clearer and facilitates analysis of the forced response and underlying mechanisms, but makes connections to realistic, transient evolution challenging. Global perturbations, on the other hand, risk conflating the effects of emissions from a given region with long-range effects from another region. Comparing studies is also challenging since

model setup and biases, and emission pathways, generally differ and can cause spurious variations between results. Hence, there is a need for a coordinated, multi-model intercomparison effort that uses (1) consistent emissions and model setup, (2) transient, realistic, aerosol perturbations following established emission scenarios, and (3) individual simulations for regions of potentially strong and rapid near-term emission changes.

In this paper, we describe the Regional Aerosol Model Intercomparison Project (RAMIP), which is designed to tackle these

challenges. RAMIP is a coordinated multi-model intercomparison project aimed at quantifying the uncertainty from aerosol-climate interactions in near-term projections. The MIP will draw on the availability of a new generation of higher-resolution models with improved representation of aerosol and related climate processes, and on the activities already ongoing in CMIP6 and its range of endorsed MIPs. Thus, RAMIP seeks to target modelers and modeling groups that previously participated in CMIP6 exercises, such as DECK (Diagnostic, Evaluation and Characterization of Klima; Eyring et al. (2016)), ScenarioMIP

(O'Neill et al., 2016), and AerChemMIP (Collins et al., 2017). Among the key scientific outcomes expected from RAMIP, we include improved knowledge of near-term hazards, of dynamical and transient responses to heterogeneous climate forcing, and of the sensitivity of near-term climate and air quality evolution to emissions policies in key aerosol-emitting countries.

Models participating in RAMIP will be used to simulate the transient climate evolution resulting from a range of plausible future emission changes in three key aerosol emission regions (South Asia, East Asia, and Africa and the Middle East), giving a

more direct link to policy decisions than global emission perturbations, and allowing for studies of aerosol transport, air quality, regionally specific climate interactions, and teleconnections and remote impacts. Generating medium-sized initial condition ensembles of simulations from each model further allows for investigations of the role of aerosols relative to internal climate variability, and projections of aerosol-induced climate forcing onto modes of variability. The main advance made possible by RAMIP is an evaluation of the full range of potential influences of realistic and regionally varied aerosol emission changes

on near-term climate evolution, as projected by current state-of-the-art Earth System Models, with fully comparable initial conditions, emissions, and experimental protocols.

In the following sections, we first document the model protocol and setup, and discuss relations to other ongoing MIPs and potential synergies from analyses combining RAMIP results with existing simulations from CMIP6. We then present a range of core findings expected from RAMIP, and how they will advance our knowledge of near-term, regional climate evolution and

risk. Finally, we show proof-of-principle results from three participating models.





## 2  Experimental design

RAMIP will explore the impact of plausible future changes in regional anthropogenic aerosol emissions within the Shared Socioeconomic Pathways (SSPs) used in ScenarioMIP and AR6 (O'Neill et al. (2016); Rao et al. (2017); Riahi et al. (2017)). The SSPs explore a wide range of global aerosol emission trajectories, from rapid decreases in emissions of carbonaceous

aerosol and sulfur dioxide in SSP1-1.9 and SSP1-2.6 to continued increases in emissions until the mid-twenty-first century in SSP3-7.0 (Rao et al. (2017); Scannell et al. (2019); Figure 1). The magnitude of the differences in global BC and $SO_2$ emissions between these SSPs are comparable to their respective increases between 1850 and 2014. Within these scenarios, there are also significant differences between emission pathways in different regions, to the extent that the sign of aerosol emission trends can differ between SSPs. Within a given region, emissions often vary strongly between scenarios. In particular, for Asia and

Africa there are large differences between the emission pathways specified in the different SSPs (Lund et al., 2019), which are likely to result in significant uncertainties in local and remote climate hazards on 30-year timescales. The core RAMIP simulations will focus on emission changes in these regions (see Figure 2).

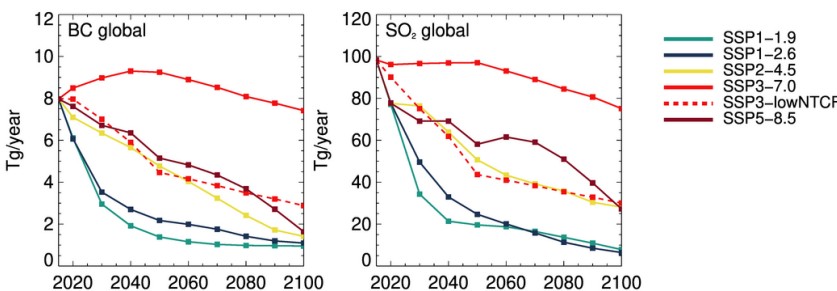

**Figure 1.** Global total emissions of (a): black carbon and (b): sulfur dioxide from a range of SSPs, including SSP3-7.0 and SSP1-2.6, upon which the RAMIP experiments are based, and the AerChemMIP SSP3-7.0-lowNTCF pathway (Table 4).

Figure 2 shows the time series of total BC and $SO_2$ emission rates over four regions from SSP3-7.0 and SSP1-2.6 from 2015 to 2100. In each case, these scenarios span the full range of aerosol and precursor emission uncertainty considered in

ScenarioMIP. For the globe and Asia, differences in the emission rate between the two scenarios reach their maximum by the mid-twenty-first century, when SSP3-7.0 emissions begin to decrease. Over Africa and the Middle East, emissions continue to increase in SSP3-7.0 until the end of the century, but growth in the difference between the two scenarios is slower in the second half of the century. RAMIP will include two sets of coupled transient experiments that will run from January 2015 to February 2051 to capture this period of rapid divergence between SSP3-7.0 and SSP1-2.6 aerosol pathways, and a set of experiments

with fixed sea surface temperatures and year 2050 emissions for the assessment of radiative forcing and rapid adjustments.

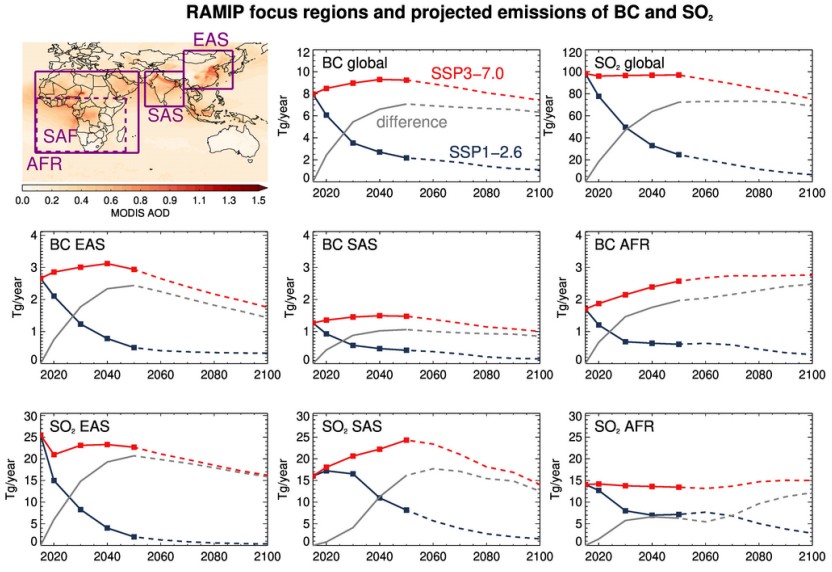

**Figure 2.** RAMIP focus regions and projected emissions of black carbon (BC) and the sulfate aerosol precursor gas sulfur dioxide ($SO_2$). The first row shows the regions where East Asian (EAS), South Asian (SAS), and African and Middle Eastern (AFR) emissions will be perturbed, and 2015 aerosol optical depth at 550nm from MODIS (combined Dark Target and Deep Blue) (Platnick, 2015), and timeseries of total global BC and $SO_2$ emission rates from SSP3-7.0 (red) and SSP1-2.6 (navy), and the difference between them (grey). The second row shows BC emission rates for the three regions, and the third row shows $SO_2$ emission rates for the three regions. East Asia is the region bounded by 95 and 133°E and 20 and 53°N, South Asia is the region bounded by 65 and 95°E and 5 and 35°N, and Africa and the Middle East is the region bounded by 20°W, 60°E, 35°S, and 35°N.

## 2.1 Transient simulations

RAMIP includes two sets of coupled transient experiments that will be used to explore the responses to regional aerosol changes (Tier 1), and the sensitivities of these responses to emission location and aerosol type (Tier 2). RAMIP transient simulations will be initialised from the end of the CMIP6 historical simulations (Eyring et al., 2016). They will use the ScenarioMIP SSP3-

7.0 simulation as a reference, as aerosol and precursor emissions continue to rise steadily in this scenario, both globally and in the focus emission regions (Figure 2). In common with all CMIP6 simulations, all RAMIP simulations will include prescribed concentrations of well-mixed greenhouse gases (GHGs) and land use changes, which will be taken from SSP3-7.0. The core, Tier 1, experiments are summarised in Table 1. These experiments use an identical setup to SSP3-7.0, but take anthropogenic aerosol and precursor emissions ($SO_2$, $SO_4$, and black and organic carbon) from SSP1-2.6 for four regions (depicted in Figure

2): globe, East Asia (EAS), South Asia (SAS), and Africa and the Middle East (AFR). In models using prescribed oxidants, these are also taken from SSP3-7.0. As such, we recommend that copies of SSP3-7.0 jobs are used as the basis for the RAMIP simulations so that only the aerosol and precursor emissions and requested output diagnostics need to be modified. An example





method for producing the regional emission files from existing SSP3-7.0 and SSP1-2.6 files is included the the Supplementary Information. The RAMIP data request is included as Appendix B.

**Table 1.** *Tier 1 transient experiments:* perturbations to regional anthropogenic aerosol and precursor emissions ($SO_2$, $SO_4$, black carbon, organic carbon). At least 10 members are requested for each experiment, intialised from the CMIP6 historical simulation. East Asia is the region bounded by 95 and 133°E and 20 and 53°N, South Asia is the region bounded by 65 and 95°E and 5 and 35°N, and Africa and the Middle East is the region bounded by 20°W and 60°E and 35°S and 35°N (Figure 2).

| Experiment | GHGs, ozone, and natural emissions | Anthropogenic aerosol emissions |
| --- | --- | --- |
| ssp370-126aer | SSP3-7.0 | SSP1-2.6 |
| ssp370-EAS126aer | SSP3-7.0 | SSP1-2.6 within the East Asia region, SSP3-7.0 otherwise |
| ssp370-SAS126aer | SSP3-7.0 | SSP1-2.6 within the South Asia region, SSP3-7.0 otherwise |
| ssp370-AFR126aer | SSP3-7.0 | SSP1-2.6 within the Africa and Middle East region, SSP3-7.0 otherwise |

Transient simulations enable us to quantify the impact of aerosol emission uncertainty on the rate of change in climate variables, and on the emergence of signals of regional climate change. Ten ensemble members are requested for both SSP3-7.0 and the RAMIP Tier 1 experiments to enable quantification of the role of internal variability and of its interaction with the forced response. Ten members is at the upper end of the ensemble size contributed by most modelling centres to the CMIP6 historical and ScenarioMIP experiments, and represents a balance between computational effort and additional information

gained per ensemble member (Monerie et al., 2022). In addition to the RAMIP experiments, participating models for which ten SSP3-7.0 ensemble members are not already available will need to run additional SSP3-7.0 ensemble members to achieve the requested ten-member SSP3-7.0 reference ensemble.

     In order to initialise the 10 ensemble members requested for the RAMIP experiments, participating models will need 10 historical ensemble members. Where it is not feasible to produce 10 members from 1850 to 2014, we recommend a procedure

similar to that used to generate the CESM2 large ensemble (Rodgers et al., 2021) whereby additional historical members (of shorter duration) are produced by branching from existing historical members in 1950, after the application of a small random perturbation to their initial atmospheric temperature fields (known as a "micro-perturbation"). Thus, the ensemble spread results from internally generated climate variability, with some sampling of internal climate variability resulting from differing ocean states.

The SSPs span a wide range of emission pathways, but cluster around three trajectories: rapid reductions until 2050, steady reductions throughout the 21st century, and sustained high emissions (Figure 1). It is unlikely that the real world will follow one of these trajectories exactly. However, it may not be sufficient to interpolate between scenarios to understand the climate response in this case, particularly when there are large differences in the regional pattern of emissions between the pathways, as is the case for Asia (Samset et al., 2019). Here, emissions are projected to increase over East and South Asia together in SSP3-

7.0, decrease together in SSP1-2.6 and SSP1-1.9, and continue the current pattern of East Asian reductions and South Asian





increases in SSP2-4.5 and SSP5-8.5. Idealised simulations have shown that the Asian climate response to aerosol changes is strongly nonlinear due to interacting atmospheric circulation responses to emission changes in neighbouring regions (Herbert et al., 2021), and there is also a suggestion of such nonlinearities in the CMIP6 ensemble (Wilcox et al., 2020). There is similar potential for nonlinearities in the response to African emission changes, where northern emission changes are dominated by

$SO_2$ and southern emission changes are dominated by BC. Dedicated experiments, included in Tier 2, are required to explore such nonlinearities fully.

The optional Tier 2 experiments, summarised in Table 2, are used to explore potential nonlinear interactions between the response to emission changes in neighbouring regions, and between the responses to particular aerosol species. SSP370-ASIA126aer is the basis for the assessment of the effect of changing emissions in both East and South Asia together. In

combination with the Tier 1 experiments, SSP370-EAS126aer and SSP370-SAS126aer, it also enables an exploration of the potential nonlinearities in the climate response that may arise when emissions in East and South Asia follow different pathways.

**Table 2.** *Tier 2 transient experiments:* exploration of regional interactions and nonlinearities. At least 10 members are requested for each experiment, intialised from the end of the CMIP6 historical simulation. East Asia is the region bounded by 95 and 133°E and 20 and 53°N, South Asia is the region bounded by 65 and 95°E and 5 and 35°N, and sub-Saharan Africa is the region bounded by 20°W and 50°E and 35°S and 12°N (Figure 2). Emissions of anthropogenic aerosol and precursor emissions ($SO_2$, $SO_4$, black carbon, organic carbon) are perturbed following SSP1-2.6 in each case for the specified region, except for ssp370-126aer_nh3nox, where $NH_3$ and $NO_x$ should also follow SSP1-2.6.

| Experiment | GHGs, ozone, and natural emissions | Anthropogenic aerosol emissions |
|---|---|---|
| ssp370-ASIA126aer | SSP3-7.0 | Anthropogenic aerosol and precursor emissions follow SSP1-2.6 within the East Asia and South Asia region, SSP3-7.0 otherwise |
| ssp370-SAF126ca | SSP3-7.0 | BC and OC emissions follow SSP1-2.6 within the sub-Saharan Africa region, and SSP3-7.0 otherwise. All other aerosol precursor emissions follow SSP3-7.0 |
| ssp370-SAS126ca | SSP3-7.0 | BC and OC emissions follow SSP1-2.6 within the South Asia region, and SSP3-7.0 otherwise. All other aerosol precursor emissions follow SSP3-7.0 |
| ssp370-126aer_nh3nox | SSP3-7.0 | Anthropogenic aerosol and precursor emissions, including $NH_3$ and $NO_x$, follow SSP1-2.6 |

Carbonaceous aerosols are likely to become a more important component of the total aerosol burden in the future (Lund et al. (2019); Samset et al. (2019)), and the climate response to changes in them is much more uncertain than the response to sulfate aerosol (Samset et al. (2016); Stjern et al. (2017)). SSP370-SAF126ca and SSP370-SAS126ca isolate the impact

of changes in carbonaceous aerosol emissions over sub-Saharan Africa and South Asia respectively. Combined with the Tier 1 experiments, SSP370-AFR126aer and SSP370-SAS126aer, they also facilitate an assessment of interactions between the responses to scattering and absorbing aerosol for the two regions.





Tier 1 and Tier 2 each require around 1500 years of coupled transient simulations. Computational requirements will vary depending on the model and its resolution (as well as additional factors, such as the high-performance computer cluster).

Indicative computational requirements are given in Appendix A for models that have been used to perform proof-of-principle simulations. As all the RAMIP experiments are projections designed to be compared to an SSP3-7.0 baseline, ten SSP3-7.0 simulations (860 years of coupled transient simulations), and thus also ten historical simulations (1650 years) and a control run (>1000 years), are required before the RAMIP simulations can be started.

## 2.2 Fixed sea surface temperature simulations

Simulations with fixed sea surface temperatures (fSST) are requested to accompany all Tier 1 experiments in order to provide additional data on forcing, rapid adjustments, and air quality impacts. These simulations will follow the RFMIP design (Pincus et al., 2016), specifying pre-industrial sea surface temperatures and sea ice concentrations, and being run for at least 30 years (Table 3, Table 4). All anthropogenic emissions will be taken from the corresponding Tier 1 experiments for the year 2050 in order to maximise the aerosol emission differences between experiments (Table 3).

**Table 3.** *Fixed-SST experiments:* perturbations to regional anthropogenic aerosol and precursor emissions. All experiments use pre-industrial SSTs, sea ice extent, and land use, following RFMIP convention (Pincus et al., 2016). Anthropogenic emissions are for the year 2050, and include the seasonal cycle. At least 30 years are requested for each experiment, and the first year is not included in analysis. East Asia is the region bounded by 95 and $133^{\circ}$E and 20 and $53^{\circ}$N, South Asia is the region bounded by 65 and $95^{\circ}$E and 5 and $35^{\circ}$N, and Africa and the Middle East is the region bounded by $20^{\circ}$W, $60^{\circ}$E, $35^{\circ}$S, and $35^{\circ}$N (Figure 2). For all experiments, 'anthropogenic aerosol emissions' are $SO_2$, $SO_4$, black carbon (BC), and organic carbon (OC). The *optional* piClim-370-126aer_nh3nox also includes perturbations to $NH_3$ and $NO_x$ emissions.

| Experiment | GHGs, ozone, and natural emissions | Anthropogenic aerosol emissions |
|---|---|---|
| piClim-370 | SSP3-7.0 | SSP3-7.0 |
| piClim-370-126aer | SSP3-7.0 | SSP1-2.6 |
| piClim-370-EAS126aer | SSP3-7.0 | SSP1-2.6 within the East Asia region, SSP3-7.0 otherwise |
| piClim-370-SAS126aer | SSP3-7.0 | SSP1-2.6 within the South Asia region, SSP3-7.0 otherwise |
| piClim-370-AFR126aer | SSP3-7.0 | SSP1-2.6 within the Africa and Middle East region, SSP3-7.0 otherwise |
| *piClim-370-126aer_nh3nox* | *SSP3-7.0* | *SSP1-2.6 $SO_2$, BC, OC, $NH_3$, and $NO_x$* |

fSST simulations are highly useful for diagnosing rapid adjustments to changes in aerosol concentrations, such as atmospheric heating profiles and lapse rates, changes to relative humidity profiles, and cloud and precipitation changes. Coupled system effects, notably SSTs and ocean-atmosphere modes of variability will subsequently impact clouds, circulation, monsoon patterns, and the precipitation response on a slower, surface-temperature dependent time scale. This will complicate the interpretations of the transient experiments, but the availability of well diagnosed ERFs and precipitation response patterns





**Table 4.** Synergies with DECK, CMIP6 historical, and other MIPs.

| MIP or project | Simulations in MIP | Simulations in RAMIP | Area of synergy |
|---|---|---|---|
| DECK[a] | piControl | All | piControl is essential for estimating internal variability. We recommend that modelling groups perform a 500-year or longer piControl run. |
| RFMIP[b] | piClim-control, piClim-aer | All fixed SST experiments | Fixed-SST experiments follow the RFMIP design, and comparison with the RFMIP piClim experiments gives context to the RAMIP fast responses. |
| CMIP6[c] | historical | All | All RAMIP experiments are initialised from CMIP6 historical experiments. |
| ScenarioMIP[d] | SSP1-2.6 and SSP3-7.0, plus additional SSPs | All | All RAMIP simulations are based on SSP1-2.6 and together with SSP3-7.0 they enable quantification of the effect of regional aerosol changes. |
| AerChemMIP[e] | SSP3-7.0-lowNTCF | SSP370-126aer | A similar scenario, with an SSP3-7.0 baseline and rapid aerosol and ozone reductions based on SSP1 air pollution legislation. |
| PDRMIP[f] | Sulasia, BCasia, Sulasired | SSP370-ASIA126aer, SSP370-SAS126ca | RAMIP transient simulations build on the idealised equilibrium experiments used in PDRMIP. |
| ISIMIP[g] | ssp370/2015soc-from-histsoc, ssp370/2015soc, ssp370/2015co2 | All transient experiments | RAMIP transient simulations can provide the GCM-based boundary conditions for ISIMIP-style experiments, All mandatory ISIMIP atmosphere variables are also requested by RAMIP. |

The associated reference papers are: [a] (Eyring et al., 2016); [b] Pincus et al. (2016); [c] (Eyring et al., 2016); [d] (O'Neill et al., 2016); [e] (Collins et al., 2017); [f] (Myhre et al., 2017b); [g] (Warszawski et al., 2014).

from rapid adjustments from fSST simulations is expected to aid in disentangling these various aspects of the response. In total, 180 years of fSST simulations are requested by RAMIP, 30 of which are optional.

Aerosol direct and indirect radiative forcing (the sum equal to the aerosol instantaneous radiative forcing) can be calculated using the method of Ghan (2013) as the difference between two simulations with the same SSTs but different aerosol and precursor gas emissions (e.g., piClim-370 and piClim-370-126aer). Direct radiative forcing is estimated as $\Delta(F - F_{clean})$, where $\Delta$ is the difference between simulations, F is the the top-of-the-atmosphere (TOA) net radiative flux, and $F_{clean}$ is the the same flux calculated as a diagnostic but neglecting scattering and absorption by individual aerosol species. The cloud radiative forcing is calculated as $\Delta(F_{clean}-F_{clear,clean})$, where $\Delta F_{clear,clean}$ is the TOA radiative flux calculated as an additional diagnostic neglecting the scattering and absorbing by both aerosols and clouds. Archival of these "double radiation call" diagnostics for





the fSST runs is encouraged for those models with this capability. Availability of model-specific instantaneous radiative forcing
for all-sky and clear-sky conditions allows a complete decomposition of rapid adjustments from the fSST simulations.

### 2.3   Optional nitrate experiments

Particulate nitrate is a significant but poorly constrained fraction of the global aerosol burden (Myhre et al., 2013). Modeling
of nitrate aerosol is made challenging by the difficulty in accurately representing the conditions for nitrate formation, and by
the strong temperature- and humidity-dependence of nitrate production and aerosol volatility.

As $SO_2$ emissions are projected to decline, the nitrate burden is projected to be increasingly important, both relatively as the
sulfate burden declines, and absolutely, as the decreasing $H_2SO_4$ burden leaves more $NH_3$ available for reaction with $HNO_3$.
Regionally, however, local decreases in $NO_x$ emissions may lead to decreased nitrate and ozone burden (Bauer et al., 2016).
The effects of modification to the sink for $NO_x$ are also uncertain, with complex implications for ozone (Bauer et al., 2007)
and OH.

The evolution of nitrate aerosol in the future, and its effect on oxidants (and hence other short-lived climate forcers) remain
largely unexplored, so far. There are important open questions regarding the role of emissions, temperature, and wet deposition
(Szopa et al., 2021) on nitrate aerosol levels in the future, with its contribution to regional air quality being unknown, given
that nitrate aerosol was excluded from future estimates of PM2.5 in AR6 due to lack of data. The evolution of nitrate loading
is further complicated by the sensitivity of aerosol to biomass burning sources of $NO_x$ (Hickman et al., 2021), which may
compensate for decreases in anthropogenic $NO_x$ emissions. A recent AerChemMIP study (Allen et al., 2021) showed large
increases in nitrate aerosol based on SSP3-7.0 (∼50% by mid-century) and moderate decreases (∼20% by mid-century) under
the mitigation pathway SSP3-7.0-lowNTCF. Moreover, these changes largely occur over two of our regions of interest, South
and East Asia.

A small number of CMIP6-generation climate models include a representation of nitrate aerosol, and RAMIP makes use
of this capability by including an optional pair of fSST and coupled transient experiments (piClim-370-126aer_nh3nox and
ssp370-126aer_nh3nox) as the basis for multi-model exploration of the drivers of future changes in nitrate aerosol loading, and
quantification of the effects of nitrate aerosol on near-future aerosol forcing and regional climate change (Tables 2 and 3). We
also request new diagnostics required to assess the nitrate budget, including the production and loss of precursor species ($NH_3$,
$NO_3$) and the ammonium nitrate ($NH_4NO_3$), and their deposition loss rates (Appendix B).

### 2.4   Diagnostics

A range of output variables across a number of temporal resolutions are requested from participating centres to enable RAMIP
experiments to be used to study the aerosol influence on a range of phenomena, including seasonal mean atmospheric and
oceanic circulation at continental scales, subseasonal evolution of the monsoons, and climate hazards including temperature
and precipitation extremes, storms, and air quality. The RAMIP data request is summarised in Appendix B. The requested
output variables for all RAMIP experiments are listed in Table B1. This core output will be CMORized and made available for
community analysis via the Centre for Environmental Data Analysis (CEDA).



Most diagnostics requested by RAMIP were included in the CMIP6 data request. However, a small number of variables have been defined specifically for RAMIP. Most of these variables are based on existing CMIP6 variables, but have been modified either to reduce their vertical resolution to enable a larger number of daily variables to be archived, or to extend them to new
chemical species to enable analysis of the nitrate budget. These variables are highlighted in Table B1 and defined in Table B2. We also request, where possible, cloud condensation nuclei at supersaturations of 0.02% and 1%, following Fanourgakis et al. (2019).

## 2.5 Relations to other MIPs

RAMIP is designed around existing experiments from CMIP6 and its endorsed MIPs, and builds on the coupled equilibrium
experiments performed in PDRMIP (Table 4). Transient simulations will be initialised from the end of the CMIP6 historical simulations, and the SSP3-7.0 experiment from ScenarioMIP will be used as the reference simulation, following AerChemMIP. Due to the short time horizon and regional focus of the RAMIP experiments, at least 10 ensemble members per experiment are requested. As such, 10 member ensembles will also be required for the historical (used to initialise the RAMIP simulations) and SSP3-7.0 simulations (the RAMIP reference case) for participating models, which many modelling centres have already
produced as part of their contribution to CMIP6.

SSP3-7.0, in which aerosol emissions continue to increase until the mid twenty-first century, is a natural reference experiment for RAMIP, as aerosol and precursor emissions decline in the other SSPs over this period (Figure 2, Figure 1). The Tier 1 SSP370-126aer is similar to the AerChemMIP SSP370-lowNTCF (Collins et al. (2017)), in which all emissions follow SSP3-7.0 except for global emissions of near-term climate forcers (methane, tropospheric ozone and its precursors, tropospheric
aerosols and their precursors, nitrous oxide, and ozone-depleting halocarbons), which are rapidly reduced following a dedicated pathway based on SSP1 (Gidden et al. (2019); Table 4). However, the magnitude of the aerosol reduction in SSP370-lowNTCF is only 50-60% as large as that in SSP370-126aer by 2050 (Figure 1).

RAMIP experiments will use the emission pathways from SSP3-7.0 and SSP1-2.6, as used in ScenarioMIP, to explore the effect of regional aerosol changes. SSP370-126aer isolates the role of global aerosol changes, and the remaining Tier 1 and 2
experiments explore the effects of emission location and aerosol type. Analysis of the Tier 1 experiments, SSP370-EAS126aer and SSP370-SAS126aer, and the Tier 2 experiment, SSP370-ASIA126aer, will also add to our understanding of the climate changes seen in SSP2-4.5 and SSP5-8.5, where emissions in East Asia decline in the early twenty-first century, while they continue to increase in South Asia (Samset et al., 2019). We also anticipate that the analysis of RAMIP simulations will be complemented by analysis of a range of ScenarioMIP experiments, in order to quantify the role of regional aerosol emissions
in the rate and magnitude of near-future climate changes, and the time of emergence of regional signals. The latter will also require a quantification of internal climate variability, which will draw on the piControl simulation from the CMIP6 DECK (Table 4).

RAMIP fSST experiments follow the design used in the RFMIP piClim experiments (Table 4). RAMIP will follow RFMIP and use pre-industrial sea surface temperatures and sea ice concentrations. In RFMIP, 2014 emissions were applied for specified
species in order to assess their impact over the historical period. For example, piClim-aer includes 2014 aerosol and precursor





emissions, with all other forcing set to 1850 values. Comparison to the piClim-control experiment, where all forcings are set to 1850 values, shows the impact of historical aerosol changes. The equivalent RAMIP experiments focus instead on their potential future impact through the comparison of simulations with 2050 emissions from different scenarios (Table 3). The RAMIP fSST experiments can also be compared directly to the RFMIP piClim-control to show the fast response to the
emission changes between 1850 and 2050.

Beyond CMIP6, RAMIP will also build on the coupled equilibrium experiments performed as part of PDRMIP (Myhre et al., 2017b) and by Westervelt et al. (2017, 2018, and 2020a). The large, idealised aerosol perturbations (a multiplication or a total removal, respectively), and long coupled equilibrium simulations used in these studies resulted in valuable information about the forced climate response to regional aerosol changes, insights into the mechanisms underpinning the responses, and
the sensitivity of the response to the model choice. RAMIP now offers a straightforward comparison of the climate response to regional aerosol changes and the response to total anthropogenic forcing on timescales relevant to climate mitigation and adaptation, which was not possible when using equilibrium experiments.

## 3    RAMIP core goals and analyses

The main goal of RAMIP is to quantify robust regional climate responses to near-future emissions of anthropogenic aerosols,
near to, and remotely from, the origin of the emissions. In order to achieve this, and going beyond reporting multi-model ensemble-mean responses, secondary goals of RAMIP include (i) quantifying the model diversity in regional biases and responses in physical aerosol-climate processes, and (ii) identifying and quantifying the interactions of regional aerosols with dynamical modes of variability in the atmosphere and ocean. Finally, if Tier 2 simulations become available, RAMIP aims to perform these investigations separately for changes in scattering and absorbing aerosols, which produce very distinct effects
on precipitation and circulation, to quantify the potential impact of nitrate aerosol in the coming decades, and to explore the interactions between climate responses to emission changes in multiple regions. By using realistic, time-varying emission perturbations based on the SSPs, RAMIP offers a straightforward comparison to existing projections, and a real-world application, that can be challenging to draw from idealised simulations. Experiments with regional emission perturbations enable a direct link to be made between regional policy decisions and regional climate impacts.
RAMIP will quantify the transient evolution of core climate determinants such as temperature, precipitation, cloud fraction and humidity; of variability indicators such as the diurnal temperature range and seasonality; and indicators of change such as extreme event intensities and occurrence rates, and the variability of daily weather. These will be interpreted in light of forcing calculations (top-of-atmosphere, surface and atmospheric) from fixed-SST simulations. The availability of a 10-member ensemble for each experiment from each model will provide a unique opportunity to separate the influence of internal variability
within one model from the inter-model differences introduced by distinct climatologies, physical process representation, and responses to forcing. Core metrics will be regional rates of change under high or low near-term aerosol emission changes, as well as changes in probability density functions of daily weather. RAMIP will also focus on transient evolutions of aerosol-cloud interactions over the regions of study, on aerosol transport and links to air pollution, and on atmospheric teleconnections





(e.g. through influences on the Walker circulation) and ocean circulation and variability changes (such as the Indian Ocean
Dipole and Atlantic Meridional Overturning Circulation).

The RAMIP output protocols include sufficient fields, and on sufficient resolutions, to disentangle the processes underlying
the simulated responses to aerosol emission changes in individual models (Appendix B), including seasonal-mean changes;
daily temperature, precipitation, and air quality extremes; and storms. These output fields will also enable assessment of the
influence of regional aerosol emissions on climate impacts, both via facilitating the computation of Climate Impact Drivers
(Ranasinghe et al. (2021)), such as extreme heat-humidity events, flooding, or fire weather, and scrutiny of the level of model
consensus on these signals, and by using them to drive climate impact models within the Inter-Sectoral Impact Model In-
tercomparison Project (ISIMIP; Warszawski et al. (2014); Table 4). RAMIP will be uniquely positioned to investigate these
processes in a multi-model setting, and to separate physical responses from internal variability.

RAMIP outputs will also enable the study of Earth System responses to future anthropogenic aerosol changes, such as
natural aerosol feedbacks, cryosphere changes, and changes in atmospheric chemistry, in models that simulate them. Dust
feedbacks in particular, may be important. The RAMIP emission regions contain several large dust source regions, and there
is some evidence that global dust burdens will increase in future (Allen et al. (2016); Tegen and Schepanski (2018)). However,
any such changes will be strongly dependent on changes in precipitation and the atmospheric circulation, making the effect of
dust feedbacks uncertain (Allen et al. (2016); Kok et al. (2018)). RAMIP simulations may also be useful for studies of future
air quality that may seek to quantify potential air quality and health improvements from specific emissions pathways, including
the air quality and health improvements due to regional emission changes.

## 4 A first look at rapid adjustments in RAMIP

Global differences in emissions of BC and $SO_2$ between SSP3-7.0 and SSP1-2.6 in 2050 are comparable in size to their
respective increases over the historical period (2014 vs. 1850). The regional emission perturbations in the Tier 1 experiments
account for between 10% and 40% of the global emission differences in 2050, and, as expected, result in only small global
mean forcing (Table 5). While earlier coupled transient experiments with Asian aerosol perturbations have shown large regional
forcing and significant and robust responses (e.g. Chen et al. (2019); Wilcox et al. (2019); Luo et al. (2020)), a possible concern
with the RAMIP design is that the small global mean forcings will lead to responses that are difficult to detect in the 36-year
transient experiments.

To preempt this concern, fixed SST experiments have already been performed with three participating RAMIP models
(Table 5), which enable the diagnosis of the effective radiative forcing (ERF) and the fast response to the changes in regional
aerosol emissions (Table 3), and provide an indication of the general model response that might be expected in the coupled
experiments. The models were run for 30 years, and the following analysis is based on all years except the first. The three
models used here include one with with a relatively strong historical aerosol ERF (CESM2, ERF = -1.37 W m$^{-2}$), one with a
relatively weak historical ERF (GFDL-CM4, ERF = -0.73 W m$^{-2}$), and one with an ERF close to the CMIP6 mean historical
aerosol ERF of -1.12 W m$^{-2}$ (UKESM1-0-LL, ERF = -1.11 W m$^{-2}$). The three models span the range of ERFs from models



**Table 5.** Models used to perform the test simulations shown in this work, their historical and future ERF, and references for RFMIP and piControl data used in this work. Historical ERFs [W m$^{-2}$] are calculated as the difference between the RFMIP simulations piClim-aer and piClim-control and quantify the response to the increase in global aerosol emissions between 1850 and 2014. 2050 ERFs [W m$^{-2}$] are global-mean values calculated as the difference between piClim-370 and piClim-370-126aer, piClim-370-AFR12aer, piClim-370-EAS126aer, and piClim-370-SAS126aer to quantify the effect of potential aerosol reductions in 2050. The spatial pattern of these forcings is shown in Figure 3 for CESM2.

| Center | Model | Historical ERF [W m$^{-2}$] | 2050 ERF [W m$^{-2}$] | | | | Model and data references |
| | | | 126aer | AFR126aer | EAS126aer | SAS126aer | |
| --- | --- | --- | --- | --- | --- | --- | --- |
| NCAR | CESM2 | -1.37 | 1.2 | 0.13 | 0.21 | 0.12 | Danabasoglu et al. (2020) |
| | | | | | | | Danabasoglu et al. (2019) |
| | | | | | | | Danabasoglu (2019) |
| NOAA-GFDL | GFDL-CM4 | -0.73 | 0.48 | 0.02 | 0.06 | 0.05 | Held et al. (2019) |
| | | | | | | | Paynter et al. (2018) |
| | | | | | | | Guo et al. (2018) |
| MOHC | UKESM1 | -1.1 | 0.56 | -0.04 | 0.08 | 0.09 | Sellar et al. (2019) |
| | | | | | | | O'Connor et al. (2019) |
| | | | | | | | Tang et al. (2019) |

anticipated to participate in RAMIP, and all lie within the 68% confidence interval of the most recent estimate of aerosol ERF (Bellouin et al., 2020). Model climatologies compared to the 5th ECMWF reanalysis (ERA5, Hersbach et al. (2020)) are briefly discussed in Appendix A, and shown in Figures A1- A3.

The global-mean ERF due to global differences in aerosol emissions in 2050 between SSP3-7.0 and SSP1-2.6 have a magnitude of between 50% and 88% of the historical aerosol forcing for the three models, with the aerosol reductions in SSP1-2.6 relative to SSP3-7.0 leading to a positive ERF of up to 1.2 W m$^{-2}$ (Table 5). Regional aerosol changes over Africa, East Asia, and South Asia, result in global-mean ERFs that are an order of magnitude smaller (Table 5, Figure 3). Although such global-mean forcings are unlikely to result in a detectable global response, the ERF is much larger near the emission region

in each case (Figure 3), and comparable to the magnitude of the historical forcing over Africa and South Asia. Such strongly heterogeneous forcing patterns can produce strong localised climate responses through their direct impact on the local radiation balance (Dong et al., 2014) and processes such as cloud and precipitation formation (Dong et al., 2019), and by inducing changes in the circulation of the atmosphere (Dong et al. (2016); Monerie et al. (2022); Wang et al. (2020)) and the ocean (Menary et al. (2020); Allen et al. (2019b)).

As the RAMIP emission regions, and some of the main anticipated response regions, experience large seasonal variations in precipitation, we show the June to August (JJA) mean response to capture the Northern Hemisphere summer monsoon season as an example of the potential climate responses in the RAMIP experiments. Annual mean anomalies largely reflect the JJA response (not shown). Figure 4 shows the JJA mean anomaly in clear-sky downwelling shortwave radiation at the

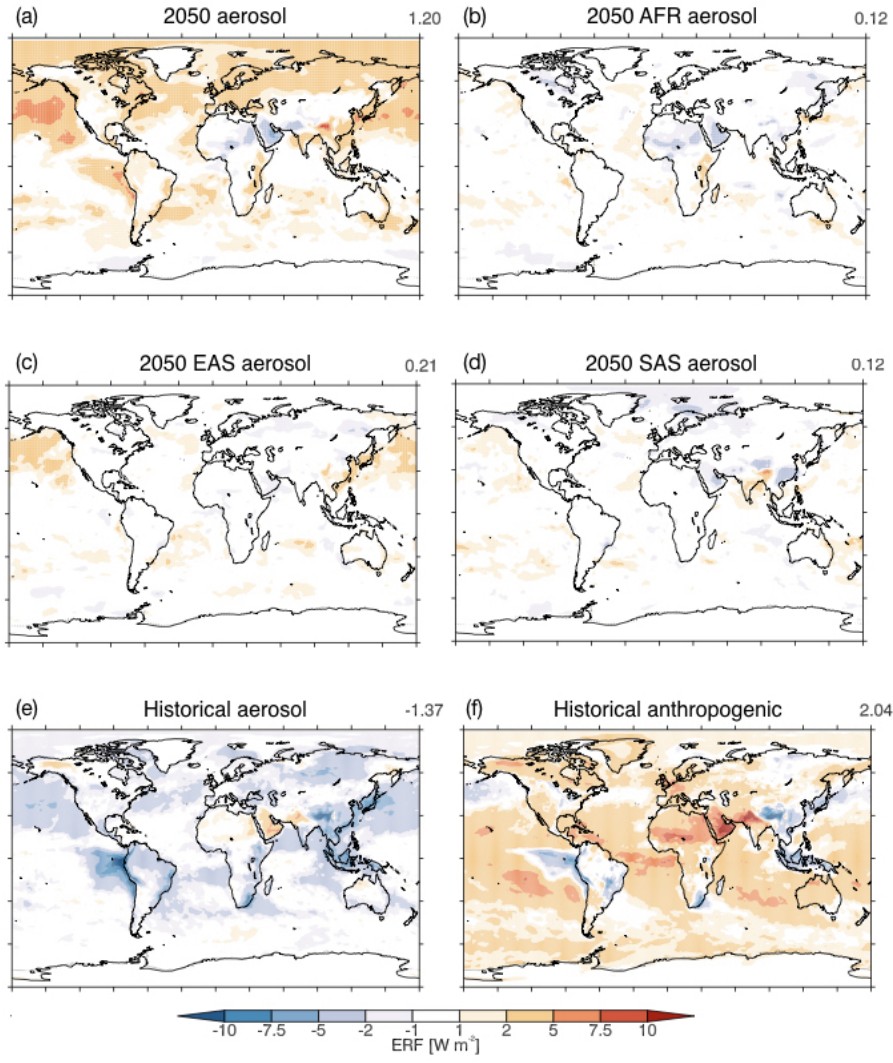

**Figure 3.** Annual mean Effective Radiative Forcing (ERF) from CESM2 for (a): piClim-370-126aer; (b): piClim-370-AFR126aer; (c): piClim-370-EAS126aer; and (d): piClim-370-SAS126aer relative to piClim-370. Historical ERF is shown on the bottom row, using the same colour scale, for (e): piClim-aer; and (f): piClim-anthro relative to piClim-control, calculated from RFMIP data. The global mean ERF [W m$^{-2}$] is shown in the top right corner of each panel.

surface for each of the perturbed aerosol fixed SST simulations relative to the control experiment, piClim-370 (Table 3), from

CESM2, GFDL-CM4, and UKESM1-0-LL. In all cases, a large, local increase in downwelling shortwave is seen due to the reduction in aerosol emissions. Increases are also seen in downstream regions resulting from the modulation of radiation by transported aerosol. The three regional experiments, piClim-370-AFR126aer, piClim-370-EAS126aer, and piClim-370-



SAS126aer, capture most of the large ($> 3$ W m$^{-2}$) anomalies seen in the global experiment, and a large fraction of the North Pacific anomaly, which is primarily the result of East Asian emission changes (Figure 4).

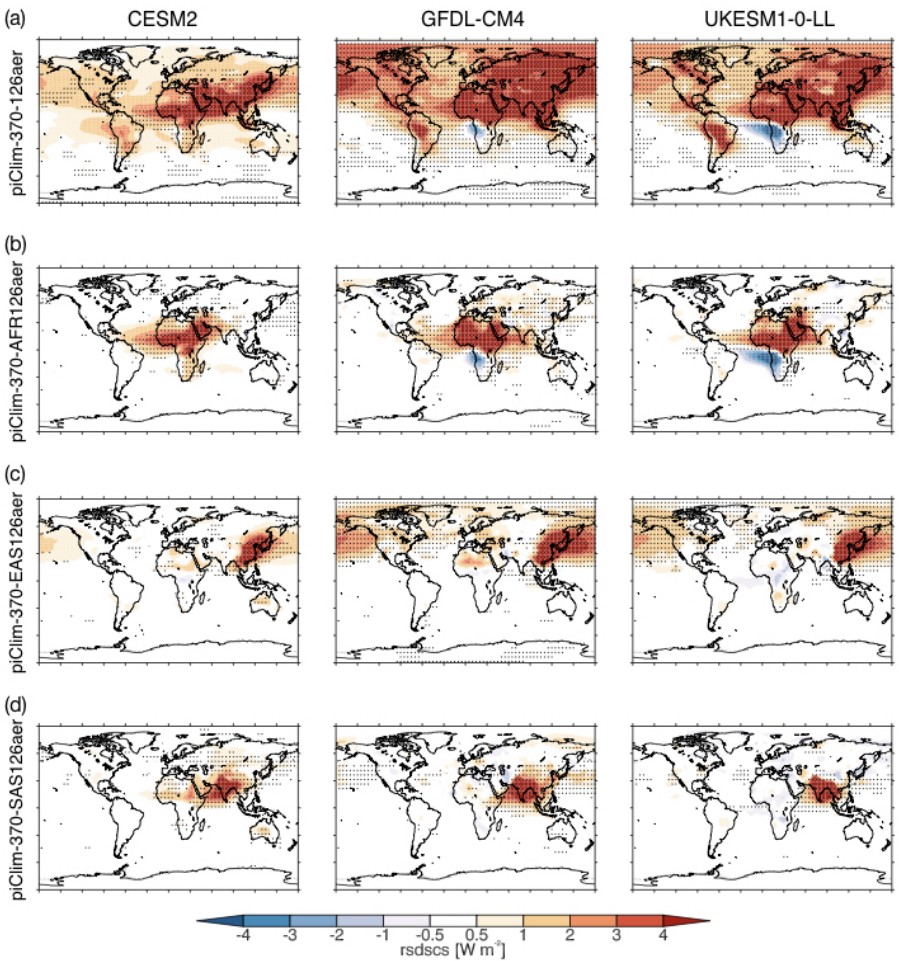

**Figure 4.** Anomalies in June-August mean downwelling shortwave radiation (clear-sky) at the surface (rsdscs) for (a): piClim-370-126aer; (b): piClim370-AFR126aer; (c): piClim-370-EAS126aer; and (d): piClim-370-SAS126aer relative to piClim-370 for CESM2, GFDL-CM4, and UKESM1-0-LL. Stippling indicates where the magnitude of the anomalies is larger than 0.5 times the interannual standard deviation.

Although the models are driven by the same emissions, inter-model differences in the response can be seen in the downwelling shortwave anomalies, which reflect differences in e.g. aerosol transport, atmospheric lifetime, and radiative properties (Figure 4). GFDL-CM4 simulates a larger increase in clear-sky downwelling shortwave radiation over the North Pacific in the two Asian experiments compared to UKESM1-0-LL and CESM2, while CESM2 simulates significant increases over the Arabian peninsula and North Africa. In piClim-370-126aer and piClim-370-AFR126aer, both GFDL-CM4 and UKESM1-0-LL

simulate large decreases in downwelling shortwave radiation relative to piClim-370, which are not seen in CESM2, reflecting the different aerosol properties in the three models. Such differences in the pattern of the radiative response to regional emission

perturbations may influence the dynamical response, and demonstrate the need for RAMIP's consistent multi-model approach to be able to identify robust responses to regional aerosol changes.

CESM2, GFDL-CM4, and UKESM1-0-LL all simulate large Asian and African precipitation anomalies in each of the
fixed SST experiments (Figure 5). A number of features are consistent between CESM2 and GFDL-CM4, including increased precipitation over east and west Africa and China in response to global aerosol reductions in SSP1-2.6, and drying over India. In all models, comparison of the SSP370-126aer and SSP370-AFR126aer anomalies suggests that the response to local aerosol changes dominates the fast African precipitation response to global aerosol reductions, while the Asian summer monsoon response is more sensitive to remote changes (Figure 5).

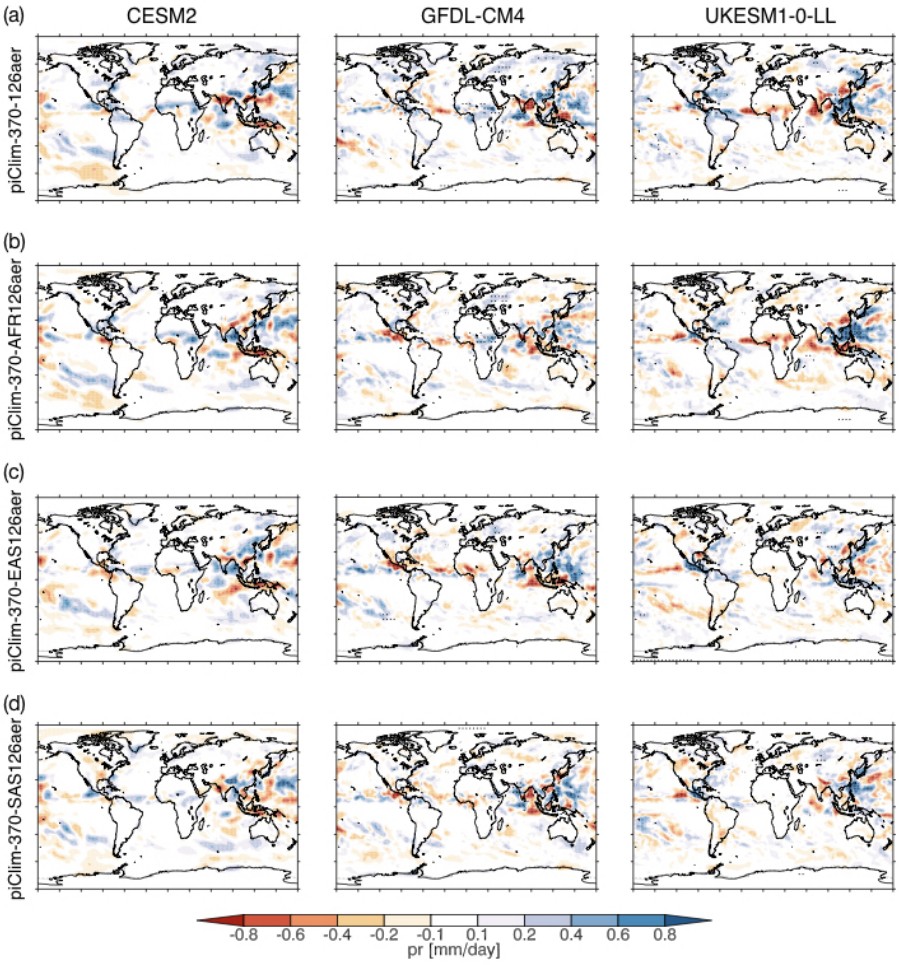

**Figure 5.** Anomalies in June-August mean precipitation (pr) for (a): piClim-370-126aer; (b): piClim370-AFR126aer; (c): piClim-370-EAS126aer; and (d): piClim-370-SAS126aer relative to piClim-370 for CESM2, GFDL-CM4, and UKESM1-0-LL. Stippling indicates where the magnitude of the anomalies is larger than 0.5 times the interannual standard deviation.





CESM2 and GFDL-CM4 both simulate different patterns of precipitation anomalies over India and China depending on the aerosol emission source (Figure 5). UKESM1-0-LL, however, simulates similar patterns but with different magnitudes, consistent with behaviour seen in earlier versions of this model (e.g. Dong et al. (2014); Dong et al. (2016); Kasoar et al. (2018)). Such differences suggest that different cloud and radiative properties, or different dynamical mechanisms, are at play in this region in different models, possibly projecting onto and being modulated by each model's baseline climatology. The

consistent experiment design in RAMIP will enable the exploration of such differences, enable the identification of responses that are robust to model differences, and further our understanding of how the long-standing model biases in this region might affect the simulated response to forcing.

     Over Africa and East Asia, the precipitation anomaly from UKESM1-0-LL generally has the opposite sign to that from CESM2 and GFDL-CM4. Although an increase in precipitation, as simulated by the latter models, is generally the expected

response to a decrease in aerosol emissions, the decrease simulated by UKESM1-0-LL is consistent within that model's climate response. In these fixed SST experiments, UKESM1-0-LL is largely mirroring its response to historical increases in global aerosol emissions, seen in the difference between the RFMIP piClim-aer and piClim-control experiments (Figure 6). However, historical experiments are unlikely to be reliable predictors of near-future responses to regional aerosol changes in all cases, as is suggested in the comparison between historical and future ERF for the three models (Table 5, Figure 3). Aerosol forcing is

mediated by clouds, which may change in response to increasing GHG concentrations, influencing the pattern and magnitude of aerosol forcing. Furthermore, some anticipated near-future aerosol emission changes, such as those over Africa, have no historical equivalent.

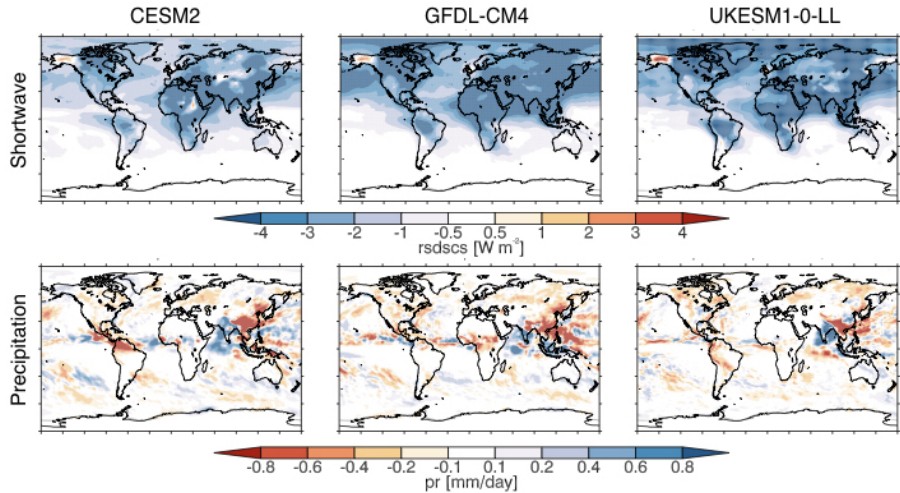

**Figure 6.** Anomalies in JJA mean (a): downwelling shortwave radiation at the surface (clear-sky); and (b): precipitation from piClim-aer relative to piClim-control for CESM2, GFDL-CM4, and UKESM1-0-LL.

     Overall, the results of these fSST test simulations are highly promising for the ability of the RAMIP simulations to help answer our core science questions. All three models show robust responses, broadly as expected from previous literature,





although with intriguing inter-model variability that illustrates the diversity in current aerosol-precipitation connections in Earth System Models and motivates the need for transient coupled simulations in a large number of models. We note, however, that the model responses in a fSST setup are markedly different to what we can expect in fully coupled simulations. Firstly, our time slice simulations were performed at, or near, the time of the largest emission differences between signal and baseline, thus maximising the ERFs. Coupled transient simulations will instead track the gradual evolution of this forcing pattern, and its subsequent climate responses. Secondly, fSST is a steady-state setup, where the models have time to equilibrate, while in transient simulations, the response time of the climate comes strongly into play. Thirdly, and most importantly, the ocean response present in coupled simulations will likely – even on the annual or decadal timescales of RAMIP – strongly modulate both the magnitude and the regional and seasonal patterns of the climate responses to aerosol emission changes. The results presented here are therefore expected to be quite different to those from the final, coupled RAMIP experiments, but can still be interpreted as a first look at the potential span of rapid adjustments to future regional aerosol changes.

## 5 Summary

Confident climate projections at regional scales are essential for better-informed adaptation and mitigation policy measures. Such predictions will require progress both on constraining radiative forcing, and in understanding the climate response to this forcing. Rapidly changing anthropogenic aerosol emissions represent a key uncertainty in such near-term projections due to the documented strong regional and global climate effects they have had over the historical period, and the large uncertainties in near-term emission pathways, radiative forcing, and the dynamical responses to heterogeneous forcing.

CMIP6 models offer significant advances in the representation of aerosol and aerosol-climate interactions compared to CMIP5 (e.g. Bellouin et al. (2013); Mulcahy et al. (2018); Kirkevåg et al. (2018); Wyser et al. (2019)). CMIP6 projections also explore a wider range of aerosol emission uncertainty than CMIP5 (Scannell et al., 2019), and many participating centres have produced several ensemble members, which better enable the study of regional climate change, and the interactions between forced and internal climate variability. RAMIP builds on the advances made in CMIP6 and its endorsed MIPs to further our understanding and quantification of the climate response to regional forcing. Focusing on the regions with the largest near-term aerosol emission uncertainty in the Shared Socioeconomic Pathways, the RAMIP Tier 1 experiments will enable the quantification of the effect of regional emission policies on changes in climate hazards local to, and remote from, the emission regions, and of their influence on regional rates of change and emergence of climate signals. Tier 2 experiments are designed to explore the interactions in the climate response to emission changes in multiple regions, and between different aerosol species. Fixed SST simulations will enable the diagnosis of aerosol forcing, and quantification of the fast response to emission changes.

There is a possibility of rapid changes in aerosol emissions over the next 30 years that are comparable in size to the increase in emissions over the historical era. The regional responses (and possible global responses via atmospheric teleconnections) to such changes are poorly understood. RAMIP will improve our physical understanding of how realistic regional aerosol emissions impact local as well as remote climate, and provide the information needed to make direct links between regional climate policies and regional climate change.



*Data availability.* Emission data for the SSPs is publicly available at the SSP database v2 (https://tntcat.iiasa.ac.at/SspDb, last access: 10 November 2022) via the "CMIP6 Emissions" tab while gridded data are available via the ESGF Input4MIPs data repository (https://esgf-node.llnl.gov/projects/input4mips/, last access: 10 November 2022).

All data requested from the RAMIP simulations described in this paper will be CMORized and distributed through the Centre for Environmental Data Analysis (CEDA) with digital object identifiers (DOIs) assigned. As in CMIP6, the model output will be freely accessible after registration. In order to document RAMIP's scientific impact and enable ongoing support of RAMIP, users are asked to acknowledge CMIP6, RAMIP, the participating modelling groups, and CEDA.

*Author contributions.* All authors designed the experiments and wrote the paper. RJA, DJP, STR, and JMK performed the simulations used in Section 4. LJW, MTL, RJA, and BHS performed the analysis.

*Competing interests.* The authors declare that they have no conflict of interest.

*Acknowledgements.* Laura J. Wilcox and Paul T. Griffiths are supported by the Natural Environment Research Council (NERC; grant NE/W004895/1, TerraFIRMA). Laura J. Wilcox, Paul T. Griffiths, James M. Keeble, and Steven T. Rumbold are additionally supported by the National Centre for Atmospheric Science. Robert J. Allen acknowledges support from NSF grant AGS-2153486. Robert J. Allen, Massimo A. Bollasina, Marianne T. Lund, Bjørn H. Samset, and Laura J. Wilcox acknowledge funding by the Research Council of Norway through grant no. 324182 (CATHY). Marianne T. Lund and Bjørn H. Samset also acknowledge funding through grant no. 248834 (QUISARC).

We acknowledge the World Climate Research Programme, which, through its Working Group on Coupled Modelling, coordinated and promoted CMIP6. We thank the climate modelling groups for producing and making available their model output, the Earth System Grid Federation (ESGF) for archiving the data and providing access, and the multiple funding agencies who support CMIP6 and ESGF. We also acknowledge the use of ERA5 data produced by ECMWF. Additional details of ERA5 can be found at https://cds.climate.copernicus.eu/. MODIS level 3 data were downloaded from the NASA Giovanni interface. The analysis in this work was performed on the JASMIN super-data-cluster (Lawrence et al., 2012). JASMIN is managed and delivered by the UK Science and Technology Facilities Council (STFC) Centre for Environmental Data Archival (CEDA).



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





## Appendix A:  Models used in fixed SST experiments

The three models used to conduct fixed SST test experiments, CESM2 (Danabasoglu et al., 2020), GFDL-CM4 (Held et al., 2019), and UKESM1-0-LL (Sellar et al., 2019), were chosen as their historical aerosol ERF spans the range of ERFs seen in the initial models participating in RAMIP, and the interquartile range of CMIP6 ERFs (-1.26 to -0.85 W m$^{-2}$). They are also independent in terms of their components, using, for example, different atmosphere and aerosol schemes.

A comparison of the 1950-2014 mean JJA mean near-surface temperature, downwelling shortwave radiation at the surface, and precipitation from the models and ERA5 is shown in Figures A1 and A2. A long period is chosen to minimise the impact of internal variability on the comparison, and JJA is shown due to the importance of the monsoon in the RAMIP emission regions. CESM2 and UKESM overestimate downwelling shortwave radiation over the Atlantic by more than 15 W m$^{-2}$ relative to ERA5 (Figures A1 and A2). UKESM has additional positive biases over Africa, the Middle East, and Asia, while CESM2 has a negative bias over Africa. All three models overestimate the downwelling shortwave radiation over Europe, to varying extents, relative to ERA5. GFDL-CM4 and UKESM1-0-LL are 1-2 K cooler than ERA5 over the Northern Hemisphere mid to high latitudes, while CESM2 is warmer (Figures A1 and A2). CESM2 and UKESM1-0-LL are both slightly warmer than ERA5 throughout the tropics.

All three models underestimate the strength of the Asian Summer Monsoon, which is a common and longstanding bias amongst climate models (Sperber et al. (2013); Wilcox et al. (2020)). UKESM1-0-LL has the largest bias over South Asia, but performs better over East Asia (Figures A1 and  A2). All models show signs of an Atlantic Intertropical Convergence Zone (ITCZ) that is located too far to the south, and a double ITCZ over the Pacific (which is also reflected in the interannual variability in the models, shown in Figure A3). In all three models, Indian ocean precipitation extends too far to the west and northwest compared to ERA5.

The model differences shown here may play a role in the diverse responses to the emission changes imposed in the RAMIP experiments. Such dependencies will be explored further when the Tier 1 experiments are available for a larger selection of models.

Initial benchmarking runs performed with these models provide indicative computational requirements for participation in RAMIP. Simulations with CESM2.1 at 1° resolution on NCAR's computer cluster "Cheyenne" require approximately 100,000 core hours per coupled transient simulation. With 10 ensemble members per experiment, and 4 (not counting the baseline) experiments for Tier 1, this equates to approximately 4M core-hours in total for the Tier 1 coupled transient experiments, and 8M core-hours for all proposed RAMIP experiments. Simulations with UKESM1-0-LL run on the Met Office Cray XC40 supercomputer require approximately 400,000 core hours per coupled transient simulation. This equates to approximately 16M core-hours in total for the Tier 1 coupled transient experiments, and 32M core-hours for all proposed RAMIP experiments.



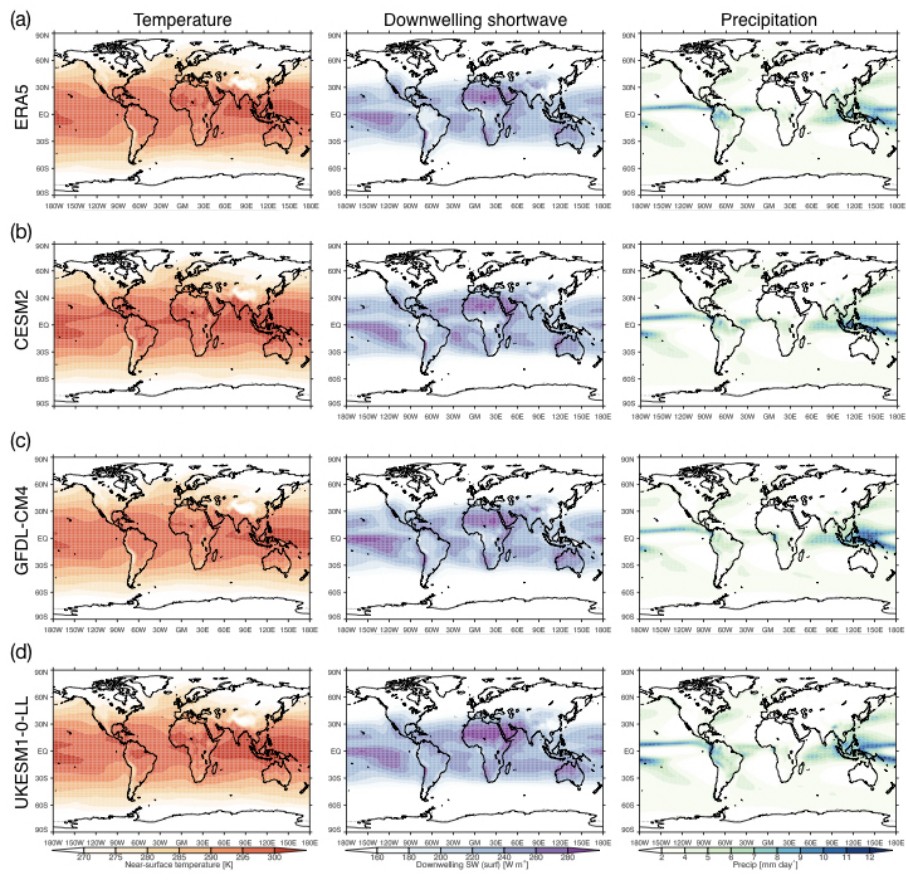

**Figure A1.** Long-term (1950-2014) mean June-August near-surface temperature, downwelling shortwave radiation at the surface, and precipitation for (a): reanalysis data from ERA5; and the three models used in the test experiments, (b): CESM2; (c): GFDL-CM4; and (d): UKESM1-0-LL, calculated using the CMIP6 historical experiment.



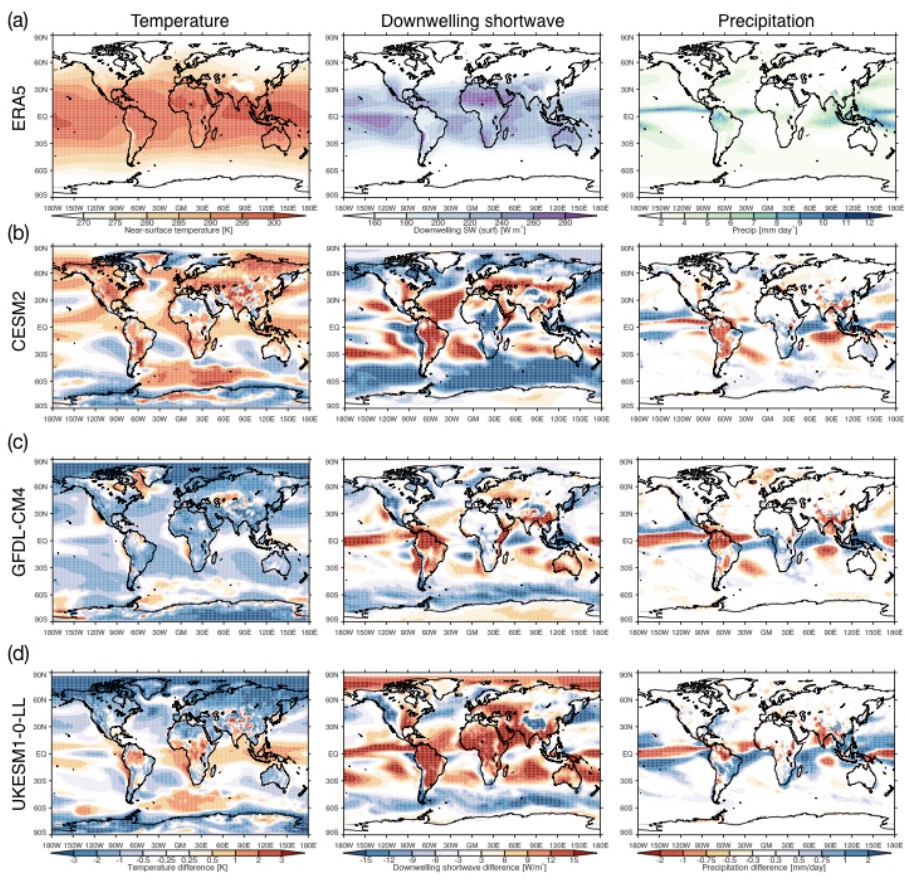

**Figure A2.** (a): 1950-2014 mean JJA near-surface temperature, downwelling shortwave radiation at the surface, and precipitation from ERA5. The remaining panels show anomalies relative to ERA5 for (b): CESM2; (c): GFDL-CM4; and (d): UKESM1-0-LL, calculated using the CMIP6 historical experiment.



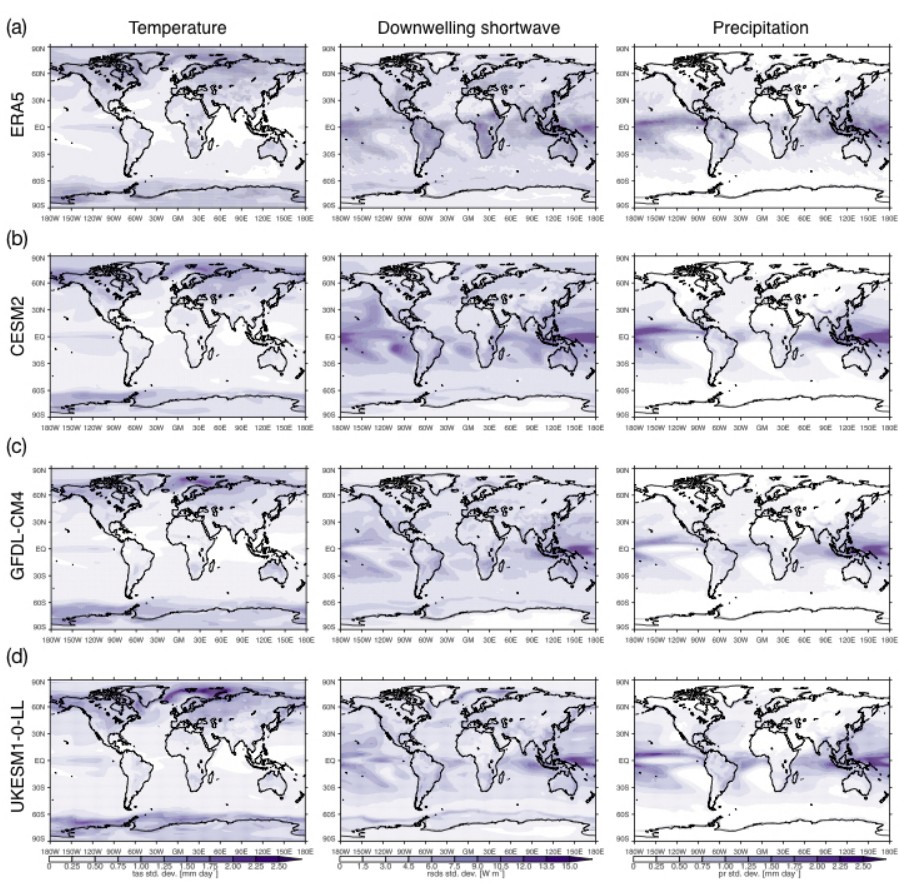

**Figure A3.** Interannual standard deviation in near-surface temperature, downwelling shortwave radiation at the surface, and precipitation for (a): ERA5; (b): CESM2; (c): GFDL-CM4; and (d): UKESM1-0-LL. For ERA5, this is calculated from detrended data for 1950 to 2020. For the CMIP6 models, this is calculated from all available years of the piControl simulation.





## Appendix B: RAMIP data request

The RAMIP data request is designed to include sufficient fields, and on sufficient temporal resolutions, to disentangle the processes underlying the simulated responses to aerosol emission changes in individual models (Tables B1 and B2), including seasonal-mean changes; daily temperature, precipitation, and air quality extremes; and storms. These output fields will also enable assessment of the influence of regional aerosol emissions on climate impacts such as extreme heat-humidity events, flooding, or fire weather. RAMIP outputs will also enable the study of Earth System responses to future anthropogenic aerosol changes, such as natural aerosol feedbacks, cryosphere changes, and changes in atmospheric chemistry, in models that simulate them.

The majority of variables requested by RAMIP are standard CMIP6 variables. However, we also request new diagnostics required to assess the nitrate budget, including the production and loss of precursor species ($NH_3$, $NO_3$) and the ammonium nitrate ($NH_4NO_3$), and their deposition loss rates. These variables are highlighted in Table B1 and defined in Table B2. We also request, where possible, cloud condensation nuclei at supersaturations of 0.02% and 1%, following Fanourgakis et al. (2019).

Where possible, we encourage the archival of double radiation call diagnostics for the fSST runs to enable a complete decomposition of rapid adjustments following the Ghan (2013) methodology. Direct radiative forcing is estimated as $\Delta$(F - $F_{clean}$), where $\Delta$ is the difference between simulations, F is the the top-of-the-atmosphere (TOA) net radiative flux, and $F_{clean}$ is the the same flux calculated as a diagnostic but neglecting scattering and absorption by individual aerosol species. The cloud radiative forcing is calculated as $\Delta$($F_{clean}$-$F_{clear,clean}$), where $\Delta F_{clear,clean}$ is the TOA radiative flux calculated as an additional diagnostic neglecting the scattering and absorbing by both aerosols and clouds.

**Table B1.** Variables requested from participating centres (CMOR variable names). The variables listed in the 'RAMIP' column are new variables designed for RAMIP, largely based on existing CMOR variables, and defined in Table B2.

| Amon | AERmon | Cfmon | Emon | Omon | RAMIPmon | Simon | AERday | CFday | day | Eday | RAMIPday | 6hrPlev |
|---|---|---|---|---|---|---|---|---|---|---|---|---|
| cl | bldep | clhcalipso | cldnvi | hfds | ccn02 | siconc | zg500 | ps | hurs | ts | hus3 | pr |
| clivi | cdnc | cllcalipso | | htovgyre | ccn1 | | | | huss | | mmrbcs | |
| clt | cheaqpso4 | clmcalipso | | htovovrt | chepnh4 | | | | pr | | mmrdusts | |
| clwvi | chegpso4 | | | mlotst | chepno3 | | | | prsn | | mmrnh4s | |
| evspsbl | drydust | | | msftmyz | dryhno3 | | | | psl | | mmrno3s | |
| hfls | drynh3 | | | sltovgyre | mmrbcs | | | | rlds | | mmroas | |
| hfss | drynh4 | | | sltovovrt | mmrdusts | | | | rsds | | mmrpm10s | |
| hurs | drynoy | | | sos | mmrnh4s | | | | sfcWind | | mmrpm2p5s | |
| hus | emianox | | | tos | mmrno3s | | | | tas | | mmrso4s | |
| mc | emidust | | | umo | mmroas | | | | tasmax | | mmrsoas | |
| pr | emilnox | | | uo | mmrpm10s | | | | tasmin | | mmrsss | |
| prsn | eminh3 | | | vmo | mmrpm2p5s | | | | | | o33 | |
| ps | eminox | | | vo | mmrso4s | | | | | | ua3 | |
| psl | o3 | | | wfo | mmrsoas | | | | | | va3 | |
| rlds | od550aer | | | wmo | mmrsss | | | | | | | |
| rldscs | od550bc | | | wo | wethno3 | | | | | | | |
| rlus | od550dust | | | | | | | | | | | |
| rlut | od550lt1aer | | | | | | | | | | | |
| rlutcs | od550oa | | | | | | | | | | | |
| rsds | od550so4 | | | | | | | | | | | |
| rsdscs | od550soa | | | | | | | | | | | |
| rsdt | reffclwtop | | | | | | | | | | | |
| rsus | wetdust | | | | | | | | | | | |
| rsuscs | wetnh3 | | | | | | | | | | | |
| rsut | wetnh4 | | | | | | | | | | | |
| rsutcs | wetnoy | | | | | | | | | | | |



**Table B2.** Monthly and daily variables defined for RAMIP. These variables are either reduced-vertical-resolution versions of existing CMOR variables, new variables required to assess the nitrate budget based on existing CMOR variables for different species, or variables used in AeroCom analysis (Fanourgakis et al., 2019).

| Variable name | Title | Relationship to existing CMOR variable |
|---|---|---|
| **Requested monthly and daily** | | |
| mmrbcs | Elemental carbon mass mixing ratio at the surface | Same as mmrbc, but reported only at the lowest model level |
| mmrdusts | Dust aerosol mass mixing ratio at the surface | Same as mmrdust, but reported only at the lowest model level |
| mmrnh4s | NH4 mass mixing ratio at the surface | Same as mmrnh4c, but reported only at the lowest model level |
| mmrno3s | NO3 aerosol mass mixing ratio at the surface | Same as mmrno3, but reported only at the lowest model level |
| mmroas | Total organic aerosol mass mixing ratio at the surface | Same as mmroas, but reported only at the lowest model level |
| mmrpm10s | PM10 mass mixing ratio at the surface | Same as mmrpm10, but reported only at the lowest model level |
| mmrpm2p5s | PM2.5 mass mixing ratio at the surface | Same as mmrpm2p5, but reported only at the lowest model level |
| mmrso4s | Aerosol sulfate mass mixing ratio at the surface | Same as mmrso4, but reported only at the lowest model level |
| mmrsoas | Secondary organic aerosol mass mixing ratio at the surface | Same as mmrsoa, but reported only at the lowest model level |
| mmrsss | Sea salt mass mixing ratio at the surface | Same as mmrss, but reported only at the lowest model level |
| **Requested monthly** | | |
| ccn02 | Cloud Condensation Nuclei concentration at 0.2% supersaturation | Following AeroCom (Fanourgakis et al., 2019) |
| ccn1 | Cloud Condensation Nuclei concentration at 1% supersaturation | Following AeroCom (Fanourgakis et al., 2019) |
| chepnh4 | Net chemical production rate of NH4 | As for chepsoa, but for NH4 |
| chepno3 | Net chemical production rate of NO3 | As for chepsoa, but for NO3 |
| dryhno3 | dry deposition rate of HNO3 | As for drynh3, but for HNO |
| wethno3 | wet deposition rate of HNO3 | As for wetbc, but for HNO3 |
| **Requested daily** | | |
| hus3 | Specific Humidity | Same as hus, but using plev3 pressure levels* |
| o33 | Ozone volume mixing ratio | Same as ua, but using plev3 pressure levels |
| ua3 | Eastward Wind | Same as ua, but using plev3 pressure levels |
| va3 | Northward Wind | Same as va, but using plev3 pressure levels |

* plev3 data is requested on three pressure levels: 850, 500, and 250 hPa.