# Peer review of "The Regional Aerosol Model Intercomparison Project (RAMIP)"

_Geoscientific Model Development, 2022_

## Referee Comment (RC2)

The manuscript presents a protocol for the Regional Aerosol Model Intercomparison Project (RAMIP). The effort is timely, as aerosol-climate interactions over different regions will emerge as a more important issue when various emission reductions will occur in the next few decades in different regions around the world. Current MIPs do not address this topic adequately. The manuscript documents the experiment designs and show some preliminary results from three models as the proof of the concept. However, I feel that some key technical details related with model physics and emission setup are missing, and some designed experiments need more justifications. Overall, the work is appropriate for GMD. I provide my specific comments below for the authors to address before I can recommend acceptance.

- To gain insight of the model spreads in predicting the regional climate impacts, we need to know how sophisticated each model treat aerosol, cloud, and radiative processes. Therefore, it would be a much valuable effort to summarize and intercompare the following aspects of the CMIP6 models in this paper: 1) how they consider mixing state of different aerosol species; 2) how they consider vertical distribution of some primary emissions, such as BC from biomass burning; 3) what cloud microphysics is being used and how CCN-cloud interactions are treated, etc. A few tables can serve this purpose. If there exists this type of synergy in the literature, please refer to them. If it is not feasible for all participating models, analyzing the three GCM mentioned in the paper should be a good starting point.

- It is unclear to me why BC and OC emissions are perturbed together in Tier 2 experiments? Are they considered to come from the same source, i.e. combustion? L218-220 mentioned the carbonaceous aerosols, but substantial amount of OA are formed via chemical processes in the populous regions like South Asia. Hence, it raises another related question: will the precursor gases of OA be perturbed in those experiments?

- It is good to see Africa is one of the target regions in RAMIP. One issue is that lots of BC are emitted by biomass burning, but burning can be either a natural process (wildfires) or anthropogenic activity. Please clarify whether and how the RAMIP emissions distinguish them.

- For the fSST experiment, pre-industrial SST and sea ice extent will be used. However, it is widely accepted that the climate responses/sensitivity hinge on the mean climate states. Therefore, we cannot expect the same aerosol impacts by running those experiment using present-day SST and SI.

- L70-73, the attribution and explanation here are way too simple. Many other factors can affect different precipitation responses to aerosol versus GHG, such as aerosol-cloud interactions. Even for absorbing aerosols like BC, the way it impacts precipitation (by altering vertical heating profile) works quite differently with GHG.

- L99-101, it has to be mentioned that for the CMIP6 models, the aerosol microphysical interactions with cloud and precipitation are considered only on the model grid scale, not the subgrid scale like shallow/deep convective cloud parameterizations. The models with explicit aerosol-cloud interactions in the subgrid cloud physics (e.g. superparameterized models) predict quite distinctive aerosol effects on cloud and precipitation (Wang et al., 2014, PNAS, "*Assessing the Effects of Anthropogenic Aerosols on Pacific Storm Track Using A Multi-Scale Global Climate Model*").

- Some important references are missing during the discussions of the importance of aerosol-climate interactions. I list two below as examples:
- L74-75, Li et al. (2017, Rev. Geophys, "*Aerosol and Monsoon Climate Interaction over Asia*") nicely summarized the state-of-the-art understanding of aerosol impacts on Asian monsoons.
- L78-85, Wang et al. (2015, JGR-Atmo, "*Atmospheric Responses to the Redistribution of Anthropogenic Aerosols*") showed the Hadley cell, jet stream, and precipitation belt responses to aerosol geospatial shifts in the past few decades in CESM1.

- Table 4, please spell out the MIP names, so people can understand what they are.

- Section 3 "RAMIP core goals and analyses" should moved forward as section 2, as readers need to know your overarching objectives before they understand your experiment design.

- Appendix A and Fig. A2, it is not appropriate to use ERA surface radiation fluxes and precipitation as truth. They are essentially predicted by the ERA modeling framework without much observational constraints.

---

## Author Response (AR1)

This manuscript describes the experiment design for RAMIP, which is designed to explore near-future responses to regional changes in aerosol emissions. Tier 1 experiments focus on emissions from regions where there is large uncertainty in future emission pathways: East Asia, South Asia, and Africa and the Middle East. Tier 2 experiments focus on the interactions between the responses to different aerosol species, and the interactions between responses to emission changes in different regions. To illustrate the expected responses to the proposed experiments, we included some preliminary results from fixed SST experiments with three models.

We now have some early results for some of the Tier 1 coupled transient experiments described in this paper. These results are encouraging, and show some interesting responses to near-future changes in regional aerosol emissions. They also show that the relatively small emission changes in Northern Hemisphere midlatitudes account for a large proportion of the midlatitude response. There are a couple of earlier studies that show that midlatitude forcing is more efficient at causing climate responses than an equivalent forcing in the tropics, so this result is not entirely unexpected, but it is very striking.

Based on our early results, the author team has decided that it would be useful to include an additional experiment with perturbed North American and European emissions in our Tier 1. The domain and emission timeseries for this new region have been added to Figure 2, and we have added the 2050 ERF to Figure 3. We show some preliminary results from this new experiment alongside those from the original experiments in Figures 4 and 5.

Beyond the above additions, changes have been made in response to the suggestions from the reviewers. We thank the reviewers for their time and constructive comments, and include point by point responses below.

**Referee 1**
This manuscript describes a model intercomparison project aimed at assess the response of climate models to near-future regional changes in aerosols. Many motivating scientific issues are described and a protocol is introduced. The protocol is based on replacing anthropogenic aerosol and precursor emissions from one scenario with another with lower emissions and performing small (10-member) ensembles of transient coupled simulations. Tier 1 simulations focus on reduced emissions globally and in three select regions; Tier 2 simulations add variants, including further emissions reductions or restricting changes to specific components. The simulations are complemented efforts to characterize the effective radiative forcing with "fixed-SST" time slice experiments at the end of the experimental period (2050). The data requested, including some new diagnostics, is described and the planned analyses are sketched out. Section 4 uses a sample of three models to demonstrate that even the relatively small aerosol perturbations are likely to translate to easily-detectable regional differences in forcing and adjustments. Connections to CMIP6-era MIPs are drawn and possible synergies highlighted.
The following comments are offered in the spirit of helping the authors make the already-fine manuscript somewhat more focused and the protocol more clear and easy to follow.

**On communication**
It it true that much of the short-term diversity in climate model projections is due to aerosols, including future emissions distributions and how each model responds to them. Nonetheless 120 lines of introduction is more than is required. It would be useful to focus the introduction more tightly on the specific questions addressed by the MIP simulations.

We completely agree, and have streamlined the introduction accordingly.

The paragraphs starting on line 200 and 260 motivate but do not describe the protocol. The paragraphs should be shortened to one or two sentences and/or placed with the rest of the motivating material.

**We have moved the paragraph starting on line 200 forward to place it with the rest of the motivating material.**

**The paragraph starting on line 260 is specific to nitrate, which we only deal with explicitly in the subsection relating to the optional nitrate experiment. As no other RAMIP experiments consider the effects of nitrate, we have kept this sentence here to avoid confusion.**

Section 4 demonstrates that changes in aerosol emissions drive different changes in radiation and precipitation across three model even in the absence of surface temperature change. The radiation changes, at least, would normally be considered part of the model-dependent forcing, not the model "response" (line 373 ff). Indeed section 4 might be better title "diversity in forcing and precipitation response" or similar.

**L373. "While earlier coupled transient experiments with Asian aerosol perturbations have shown large regional forcing and significant and robust responses (e.g. Chen et al. (2019); Wilcox et al. (2019); Luo et al. (2020)), a possible concern with the RAMIP design is that the small global mean forcings will lead to responses that are difficult to detect in the 36-year transient experiments."**

**This is referring to coupled experiments, which yield the total climate response. Typically, this total climate response can be decomposed into a "fast" and "slow" response, with the fast response–otherwise known as the rapid adjustments–based on fSST experiments (i.e., fixed SSTs) and the slow response (due to changes in SSTs) estimated as the total minus the fast response. In this section, we use the term "fast response", as it shows results from fSST experiments. Nonetheless, we have changed the title of this section to "A first look at Effective Radiative Forcing and fixed SST responses in RAMIP". This better encompasses the diversity of fSST results discussed in this section, which include ERF and precipitation (as noted by the reviewer), but also downwelling surface clear-sky solar radiation.**

**On the protocol**
It would be useful to be explicit about the time window the simulations in multiple places, including in the table captions.

**All requested transient coupled simulations span 01/2015-02/2051. The fSST experiments are 30 years long. We have added this information to the table captions.**

It would be useful to have a small table detailing the boundaries of the regions so reader don't need to read the captions closely.

**Figure 2a graphically illustrates the regions. Region boundaries are also listed in the corresponding caption (and several additional captions). We worry that adding another table would harm the readability of the manuscript without adding much value, so we have instead added a note to the main text explicitly referring the reader to the caption of Figure 2 for the definition of the region boundaries.**

Asking participants to create their own emissions files (Section 2.1, Supplemental Information) seems like an unnecessary opportunity for mistakes to be introduced. Could the MIP not create and distribute these files?

**We understand the reviewer's concern. However, it is not possible to create such standardised emission files that will be of universal use to the various models participating in RAMIP. Each model has a unique emission file standard and structure. CESM2 has 22 different emission files that need to be modified while UKESM has 12, for example.**

**To moderate the risk of errors being introduced via the emission files, we have provided an example script that can be used to modify existing SSP emission files to create the regional combinations required by RAMIP (included in the Supplementary Information). Straightforward and standard steps (e.g. spatial trend maps, time series, etc.) can be adopted to test the emissions files before performing coupled simulations. Participants may also choose to first perform the fSST experiments to further evaluate the modified emission files, before performing the more computationally intensive coupled experiments. We have added these recommendations.**

The MIP proposers might explain why they have adopted the RFMIP time-slice protocol rather than the protocol for computing transient ERF (3-member, fixed-SST), given the small cost relative to the coupled model experiments and the strong time evolution of the aerosol emissions.

**Transient ERF simulations are less computationally expensive than the coupled runs, but not by very much. The atmospheric component of most models likely to participate in RAMIP consumes most of the computational requirements (e.g. UKESM coupled RAMIP simulations use 576 processors for the atmosphere and only 108 processors for the ocean). Relative to the 30-year fSST time-slice ERF approach, transient ERF simulations are considerably more expensive (about 3x more expensive based on 30 years versus ~110 years). Ultimately, we are interested in the transient coupled response. The fSST experiments are meant to help understand the ERF and rapid adjustments. Such goals can be accomplished through the fSST time-slice approach.**

**On graphical communication**
Figure 1 might replace the two right-most panels in figure 2.

**Thank you for the suggestion, but we think it is best to retain the current format. The two panels in Figure 1 are not easily transferable to Figure 2, and the two figures are meant to emphasise different points. Figure 1 shows the diversity in global BC and SO2 emission pathways across 6 SSPs. Figure 1 emphasises the diversity in near-term global aerosol emissions pathways, and thus motivates RAMIP, and explains our choice of SSPs to serve as the basis for the RAMIP experiments. Figure 2, however, is explicitly showing the BC/SO2 evolution from the two scenarios (SSP3-7.0 and SSP1-2.6) chosen for RAMIP, as well as the corresponding difference, for the global mean as well as each of our regions. This is a demonstration of the RAMIP experiment design, and a visual illustration of why we choose to run the experiments to 2050 (the point of maximum difference between SSP3-7.0 and SSP1-2.6 in most regions in the Tier 1 experiments).**

Most figures contain more white space than need. In figure 2 removing repeated axis labels would be a benefit. Figures with 3x3 maps are too small to convey much information, and all the multi-map figures have loads of unnecessary space between the panels.

**We have reduced the amount of white space, and increased the panel size, in Figure 3, 4, 5, A1, A2, and A3. We have removed axis labels from Figures A1, A2, and A3. In Figures A1, A2, and A3 we have also increased the size of the colour bars, and the font size of their labels. We have removed some of the white space and x-axis labels from Figure 2 but have retained all the y-axis labels. The y-axis labels are different for each region, and we felt this wasn't as clear without the labels on each panel.**

**While we appreciate that the panel size is a little small in Figures 4 and 5, and A1-3, we think the format of these figures represents the best compromise. These figures are meant to support high-level qualitative discussion. They show large-scale differences between experiments and models, and are not intended to be used for examining small-scale features, or supporting quantitative discussion (we leave such analysis for future publications using RAMIP). We think it is most important to have the models and experiments side by side in the same figure, which**

**necessitates the use of several panels. We hope that the reviewer finds these figures acceptable now that we have used the reduction in white space to increase the size of the individual panels.**

Figure 5: "Stippling indicates"… is there any stippling? How was this metric for significance arrived at?

**The stippling was missing from the CESM2 panels in the submitted manuscript. We have fixed this issue now. We have also decreased the spacing between stipples in Figure 5 where there is not a lot of stippling so that it is easier to see.**

**The stippling is used to show where the anomalies are large relative to the interannual variability. Our aim with these figures is to illustrate the different forcings and responses we might expect from the full set of RAMIP experiments, rather than presenting a rigorous scientific analysis (which we save for future work). A generic metric based on signal-to-noise was therefore chosen to convey this point..**

**We found one occurrence in the text where we had referred to a stippled anomaly as 'significant'. This was not intentional and we have amended it accordingly.**

**Detailed comments**
Line 64: "Aerosol-precipitation interactions" would be more accurate than "aerosol-climate"
**Changed**

Lines 67-70 imply that differences in the global hydrologic sensitivity imply strong regional changes - this deserves a citation if its true.
**These lines have been removed as part of the streamlining of the introduction suggested by Reviewer 1.**

What connects the many issues raised in the paragraph starting on line 107?
**We have split this paragraph into two so that each focuses on one issue. The first deals with emission and forcing uncertainty as highlighted in AR6. The second focuses on the uncertainty related to experiment design, and discusses choices made in previous studies, and the implications of these.**

Line 128: "designed to address these challenges…" there are so many challenges listed to this point that it's hard to guess the point of the MIP.
**We have made this sentence more specific, thank you.**

Line 130: a diversity of model results does not represent "uncertainty." Perhaps this could be made more specific and concrete.
**A diversity of model results is in part related to structural uncertainty. The design of the RAMIP experiments means that the MIP is also exploring the effects of emission uncertainty. The 10-member ensembles requested will enable us to better quantify the effects of internal variability better than in previous MIPs. Tier 2 experiments will allow us to quantify some aspects of process uncertainty.**

**We have rephrased this section of the manuscript to be more specific. It now reads:**
**"The Regional Aerosol Model Intercomparison Project (RAMIP) is a coordinated multi-model intercomparison project aimed at quantifying the climate and air quality responses to changing regional emissions in near-term projections."**

Line 138: It is presumably the simulations, rather than the models making them, that will allow the more direct link described here.
**Rephrased, thank you.**

Line 176: Perhaps "In common with most CMIP6-related…"  CMIP6 encompasses only the historical simulation. Other simulations either come from the CMIP DECK or from satellite MIPs.
**Changed as suggested**

Line 181 "copies of jobs" is rather technical - could this explained more generically?
**The main target audience of the paper is modelling groups who have already participated in CMIP6. 'copies of jobs' is already quite generic.  A precise phrase would require more technical details, like restart files, namelist files, component sets, so we think we have the balance right here.**

Most of line 224 is superfluous.
**We have removed this sentence.**

Line 280: Does CEDA plan to CMOR-ize everyone's output, or are modeling centers expected to provide CMORized output? How will the CMOR tables be prepared?
**Modelling centres are expected to provide CMORized output, as for CMIP6. The participating centres will be best-placed to do this, given the unique requirements of each model. Transferring only CMORized output also reduces the amount of data that must be transferred. We have amended the sentence originally at line 280 to make clear that while CEDA will host the data, participating centres need to CMORize it.**

**CMOR tables have been produced by Laura Wilcox in consultation with Matthew Mizielinski, and are currently made available directly to participating centres. They will be made available via a RAMIP Github page when this documentation paper is finalised.**

Line 290: PDRMIP is mentioned here but not introduced or cited until later.
**Citation added**

Line 368: This begins section 4 quite abruptly. A bridging sentence would be welcome.
**Added**

Line 375: "prempt" -> "address or similar.
**This phrase was lost as part of the addition of the bridging sentence suggested in the previous comment.**

Line 375: Perhaps remove "participating" since the experiment hasn't happened yet.
**Changed as suggested**

**Referee 2**
The manuscript presents a protocol for the Regional Aerosol Model Intercomparison Project (RAMIP). The effort is timely, as aerosol-climate interactions over different regions will emerge as a more important issue when various emission reductions will occur in the next few decades in different regions around the world. Current MIPs do not address this topic adequately. The manuscript documents the experiment designs and show some preliminary results from three models as the proof of the concept. However, I feel that some key technical details related with model
 physics and emission setup are missing, and some designed experiments need more justifications. Overall, the work is appropriate for GMD. I provide my specific comments below for the authors to address before I can recommend acceptance.

 • To gain insight of the model spreads in predicting the regional climate impacts, we need to know how sophisticated each model treat aerosol, cloud, and radiative processes. Therefore, it would be a much valuable effort to summarize and intercompare the following aspects of the CMIP6 models in this paper: 1)how they consider mixing state of different aerosol species; 2) how they consider vertical distribution of some primary emissions, such as BC from biomass burning; 3) what cloud microphysics is being used and how CCN-cloud interactions are treated, etc. A few tables can serve

this purpose. If there exists this type of synergy in the literature, please refer to them. If it is not feasible for all participating models, analyzing the three GCM mentioned in the paper should be a good starting point.

**We agree that understanding how each model represents aerosol, cloud and radiative processes is important to understanding model diversity in regional climate responses due to aerosols. However, we do not think such detailed information is necessary for inclusion in a model experiment description paper. This paper is meant to motivate and describe RAMIP, outline the experimental design, and provide relatively simple proof-of-concept results (which will hopefully encourage participation by modelling groups). This introductory paper is not intended to document model parameterizations, or to try to understand response diversity. As yet, we do not know which models will participate in RAMIP, and some participating models may not be those used in CMIP6 (e.g., GFDL-SPEAR), or may include updated features/parameterizations/processes relative to their CMIP6 version (e.g. UKESM1.1 and/or UKESM2). We have, however, added a little more information (in the Appendix) on the three models used to perform the experiments shown in Section 4.**

**We completely agree with the importance of this information, but this paper is not the appropriate place to include it. Once we have formal commitments by modelling groups and experiments are completed, the first RAMIP results paper will document each model's representation of aerosols, clouds and radiative processes (and other components).**

• It is unclear to me why BC and OC emissions are perturbed together in Tier 2 experiments? Are they considered to come from the same source, i.e. combustion? L218-220 mentioned the carbonaceous aerosols, but substantial amount of OA are formed via chemical processes in the populous regions like South Asia. Hence, it raises another related question: will the precursor gases of OA be perturbed in those experiments?

**The focus of some of our Tier 2 experiments (e.g., ssp370-SAF126ca and ssp370-SAS126ca) is to isolate the regional responses due to carbonaceous aerosols. Hence, both BC and OC are perturbed together. These simulations do not include perturbations of the precursor gasses responsible for SOA. Not all CMIP6 generation models include interactive SOA, so neglecting SOA precursors makes this a cleaner experiment.**

• It is good to see Africa is one of the target regions in RAMIP. One issue is that lots of BC are emitted by biomass burning, but burning can be either a natural process (wildfires) or anthropogenic activity. Please clarify whether and how the RAMIP emissions distinguish them.

**RAMIP is based on the SSPs, which do not include specified changes in wildfires. Our experiment design considers only anthropogenic activity. However, it is possible that some models used to perform our experiments will include interactive wildfire emissions. We have added clarification that we consider only anthropogenic emissions to a few places in the manuscript.**

• For the fSST experiment, pre-industrial SST and sea ice extent will be used. However, it is widely accepted that the climate responses/sensitivity hinge on the mean climate states. Therefore, we cannot expect the same aerosol impacts by running those experiment using present-day SST and SI.

**We agree. However, the canonical definition of ERF uses 1850 as the base state, which we have adopted. Our fSST simulations are meant to diagnose ERF and rapid adjustments, which allows insight on model diversity pertaining to their aerosol ERF and the corresponding atmospheric response, including clouds.**

• L70-73, the attribution and explanation here are way too simple. Many other factors can affect different precipitation responses to aerosol versus GHG, such as aerosol-cloud interactions. Even for

absorbing aerosols like BC, the way it impacts precipitation (by altering vertical heating profile) works quite differently with GHG.

**We have removed these sentences when streamlining the introduction in response to comments from Reviewer 1.**

• L99-101, it has to be mentioned that for the CMIP6 models, the aerosol microphysical interactions with cloud and precipitation are considered only on the model grid scale, not the subgrid scale like shallow/deep convective cloud parameterizations. The models with explicit aerosol-cloud interactions in the subgrid cloud physics (e.g. superparameterized models) predict quite distinctive aerosol effects on cloud and precipitation (Wang et al., 2014, PNAS, "Assessing the Effects of Anthropogenic Aerosols on Pacific Storm Track Using A Multi-Scale Global Climate Model").

**We agree that the way aerosol-cloud interactions are represented in models is important to the simulated effects, and have now included a mention of the grid-scale representation of aerosol microphysical interactions with cloud and precipitation in this paragraph.**

• Some important references are missing during the discussions of the importance of aerosol-climate interactions. I list two below as examples:
• L74-75, Li et al. (2017, Rev. Geophys, "Aerosol and Monsoon Climate Interaction over Asia") nicely summarized the state-of-the-art understanding of aerosol impacts on Asian monsoons.
• L78-85, Wang et al. (2015, JGR-Atmo, "Atmospheric Responses to the Redistribution of Anthropogenic Aerosols") showed the Hadley cell, jet stream, and precipitation belt responses to aerosol geospatial shifts in the past few decades in CESM1.

**These references are now included.**

• Table 4, please spell out the MIP names, so people can understand what they are.

**It is common practice to refer to these MIPs by their acronyms, and we expect that these are more recognisable than their full names in all cases. We have used the table caption to spell out the MIP names (please note that latexdiff hasn't picked this up, so it's not highlighted in the marked up version of the manuscript).**

• Section 3 "RAMIP core goals and analyses" should moved forward as section 2, as readers need to know your overarching objectives before they understand your experiment design.

**We have moved the core goals to Section 2, ahead of the definition of the experiment design. We have kept the discussion of the core analyses after the introduction of the experiment design. This discussion is slightly more technical and relies on details of the experiment design. As the intended analysis informs the data request, we have wrapped this discussion into that subsection.**

• Appendix A and Fig. A2, it is not appropriate to use ERA surface radiation fluxes and precipitation as truth. They are essentially predicted by the ERA modeling framework without much observational constraints.

**While we agree that a detailed model evaluation should include observations, especially for variables that are poorly constrained in reanalyses, that is not what we are aiming to do here. The brief comparison of the three models used to perform proof of concept experiments in the Appendix is intended only to be a qualitative demonstration of the differences between models, not a comparison of the models to the 'truth'. We consider ERA5 to be sufficient for this purpose, and, for use in an Appendix, find it useful for simplifying the discussion.**

A comparison to observations here would require us to shorten the time period used due to the limited availability of surface shortwave observations, which complicates the comparison as internal variability is likely to play a more prominent role in any differences between the models and observations on such timescales. We would also need to compare the models to multiple observational datasets, as the differences between observational datasets can be as large as the differences between the models themselves (e.g. Wilcox et al., 2020; Dong et al., 2023). This would be appropriate for a science paper but feels distracting for a model experiment description paper where no quantitative analysis is performed.

Dong, B., Sutton, R.T. & Wilcox, L.J., 2023: Decadal trends in surface solar radiation and cloud cover over the North Atlantic sector during the last four decades: drivers and physical processes. *Clim Dyn* 60, 2533–2546, https://doi.org/10.1007/s00382-022-06438-3

Wilcox, L. J., Liu, Z., Samset, B. H., Hawkins, E., Lund, M. T., Nordling, K., Undorf, S., Bollasina, M., Ekman, A. M. L., Krishnan, S., Merikanto, J., and Turner, A. G., 2020: Accelerated increases in global and Asian summer monsoon precipitation from future aerosol reductions, Atmos. Chem. Phys., 20, 11955–11977, https://doi.org/10.5194/acp-20-11955-2020.

---

## Author Response (AR2)

*Referee 2 suggests that the details of the modelling of aerosols, cloud and precipitation in each model should be detailed. I agree that this is probably excessive for an experiment description paper although it would be appropriate to highlight the differences between the models used in the small proof of concept ensemble. What is missing from the manuscript, however, is a basic description of the state of the art of the modelling of these elements in current CMIP models. The introduction in the previous version of the manuscript did include some of this kind of information but it has been removed in response to Referee 1's request to shorten the introduction. Therefore I suggest that a new short section be added to include such an overview.*

We have expanded the paragraph beginning 'Firstly, …' on line 84 of the revised manuscript to include a description of the state of the art in CMIP6 models. This reinstates the information removed in response to Referee 1's request, and adds a little more detail about the modelling of aerosol-cloud interactions compared to the previous version of the manuscript.

Important differences between the models in the proof of concept ensemble, likely to affect their aerosol forcing, were added to the Appendix in response to Referee 2's comments.

*The SSP database site seems to be functional, but I could not see anything for RAMIP on the input4mips site linked to in the manuscript. Perhaps more instruction on how to obtain the files is required? The concept at GMD is that a scientist with a suitable model should be able to setup and run the experiment independently (ie. without directly contacting the authors of the paper or other members of the RAMIP community).*

RAMIP has been designed around the SSPs, which were produced for ScenarioMIP, and the relevant emissions files are listed under ScenarioMIP on the input4mips site. We've added additional instructions for obtaining these files to the data availability statement.